# Meningeal lymphatic drainage promotes T cell responses against *Toxoplasma gondii* but is dispensable for parasite control in the brain

Michael A Kovacs, Maureen N Cowan, Isaac W Babcock, Lydia A Sibley, Katherine Still, Samantha J Batista, Sydney A Labuzan, Ish Sethi, Tajie H Harris*

Center for Brain Immunology and Glia, Department of Neuroscience, University of Virginia, Charlottesville, United States

**Abstract** The discovery of meningeal lymphatic vessels that drain the CNS has prompted new insights into how immune responses develop in the brain. In this study, we examined how T cell responses against CNS-derived antigen develop in the context of infection. We found that meningeal lymphatic drainage promotes CD4+ and CD8+ T cell responses against the neurotropic parasite *Toxoplasma gondii* in mice, and we observed changes in the dendritic cell compartment of the dural meninges that may support this process. Indeed, we found that mice chronically, but not acutely, infected with *T. gondii* exhibited a significant expansion and activation of type 1 and type 2 conventional dendritic cells (cDC) in the dural meninges. cDC1s and cDC2s were both capable of sampling cerebrospinal fluid (CSF)-derived protein and were found to harbor processed CSF-derived protein in the draining deep cervical lymph nodes. Disrupting meningeal lymphatic drainage via ligation surgery led to a reduction in CD103+ cDC1 and cDC2 number in the deep cervical lymph nodes and caused an impairment in cDC1 and cDC2 maturation. Concomitantly, lymphatic vessel ligation impaired CD4+ and CD8+ T cell activation, proliferation, and IFN-γ production at this site. Surprisingly, however, parasite-specific T cell responses in the brain remained intact following ligation, which may be due to concurrent activation of T cells at non-CNS-draining sites during chronic infection. Collectively, our work reveals that CNS lymphatic drainage supports the development of peripheral T cell responses against *T. gondii* but remains dispensable for immune protection of the brain.

## Editor's evaluation

Kovacs et al. provide important and valuable data on the role of meningeal lymphatic drainage in T cell responses during chronic *Toxoplasma gondii* (T.g.) infection in mice. They present compelling data showing dendritic cell (DC) accumulation in the dura and CSF at 6 weeks post-infection, which matches with the replication peak of T.g. in the brain, and with T cell expansion/activation in the draining lymph node (dCLN). They also convincingly show that during chronic infection, antigen-specific T cells are generated not only in the dCLN but also in the periphery (ILN), which could account for the presence of T cells in the brain after surgical blockade of the lymphatics. This study highlights also how the CNS is protected against environmental challenges.

*For correspondence:
tajieharris@virginia.edu

Competing interest: The authors declare that no competing interests exist.

## Introduction

Although the presence of T cells in the CNS can be pathological, T cells play an essential role in orchestrating host-protective immune responses in the CNS during infection (*Ellwardt et al., 2016*; *Korn and Kallies, 2017*). At present, the biological mechanisms that drive the formation of T cell responses in the CNS remain poorly defined. One well-studied brain-tropic pathogen is *Toxoplasma gondii*, an intracellular protozoan parasite that causes chronic, lifelong infection in a wide range of mammalian hosts, including humans (*Hill and Dubey, 2002*; *Pappas et al., 2009*). Although the course of infection is typically asymptomatic in humans, severe neurologic disease can manifest in immuno-compromised individuals, including HIV/AIDS patients (*Luft and Remington, 1992*). Consistent with this susceptibility, experimental studies have demonstrated that mice deficient in T cell responses are unable to protect the host from fatal disease (*Gazzinelli et al., 1992*). For this reason, murine infection with *T. gondii* is a useful model for investigating how T cell responses develop in the CNS.

At homeostasis, the immunologically quiescent brain harbors very few T cells, but in response to *T. gondii* infection, parasite-specific T cells are actively and continuously recruited to the brain (*Harris et al., 2012*). The process is tightly regulated, as circulating T cells that have been activated in the periphery can only persist within the brain when their cognate antigen is expressed in the brain (*Schaeffer et al., 2009*; *Wilson et al., 2009*). Disrupting entry of newly activated T cells into the brain leads to impaired parasite restriction, indicating that ongoing T cell stimulation in the periphery is essential (*Harris et al., 2012*; *Wilson et al., 2009*). However, because the brain parenchyma does not harbor lymphatic vessels, it remains poorly understood how peripheral T cells become alerted to the presence of microbial antigen in the brain.

In other organs, lymphatic vessels serve as conduits for the transport of tissue-derived antigen and dendritic cells to lymph nodes, where naive and memory T cells are optimally positioned for detection of their cognate antigen (*Thomas et al., 2016*; *Gasteiger et al., 2016*). The recent discovery of functional lymphatic vessels in the dura mater layer of meninges has prompted a significant reconsideration of how the CNS engages the peripheral immune system (*Louveau et al., 2015b*; *Aspelund et al., 2015*). Meningeal lymphatic vessels are observed in rodents, primates, and humans (*Absinta et al., 2017*; *Albayram et al., 2022*), and in experimental models of brain cancer and autoimmunity these vessels have been shown to play an integral role in regulating T cell responses in the CNS (*Song et al., 2020*; *Louveau et al., 2018b*). Mouse studies have demonstrated that meningeal lymphatic vessels convey macromolecules and immune cells from the meninges and cerebrospinal fluid (CSF) to the deep cervical lymph nodes (*Louveau et al., 2018b*). Indeed, when model antigens like ovalbumin (OVA) are injected into the brain, these molecules travel from the brain interstitium into the CSF via glymphatic flow (*Iliff et al., 2012*) and have the potential to be presented to T cells in the deep cervical lymph nodes (*Ling et al., 2003*; *Harris et al., 2014*).

In a murine model of acute brain infection, it was shown that meningeal lymphatic drainage contributes to the clearance of Japanese encephalitis virus from the brain and promotes host survival (*Li et al., 2022*). However, questions remain regarding how the meningeal lymphatic system affects T cell responses against brain-derived microbial antigen and the specific role that the meninges and meningeal lymphatic drainage play during chronic brain infection. In this study, we report an expansion of type 1 and type 2 conventional dendritic cell (cDC) populations in the dural meninges following infection with *T. gondii*. cDC2s in particular displayed broad upregulation of co-stimulatory molecules and MHC class II, while both cDC1s and cDC2s were capable of sampling CSF-derived protein. Lymphatic vessel ligation experiments revealed that meningeal lymphatic drainage is required for dendritic cell responses in the deep cervical lymph nodes and promotes robust T cell activation, proliferation, and cytokine production at this site. However, in contrast to the finding that meningeal lymphatic drainage is host-protective during acute viral infection of the brain (*Li et al., 2022*), we observed that meningeal lymphatic drainage is dispensable for controlling *T. gondii* infection of the brain, with the T cell response in the brain remaining intact following surgical disruption of meningeal lymphatic outflow. Concurrent activation of T cells at alternative sites, including lymph nodes that do not drain the CNS, potentially explains the durability of the T cell response in the brain. Overall, our findings highlight the role of the dural meninges in the immune response to chronic brain infection, providing new evidence that meningeal lymphatic drainage promotes T cell responses against antigen expressed in the brain.

# Results

## Expansion of type 1 and type 2 conventional dendritic cell populations in the dural meninges during chronic brain infection

The dura mater layer of meninges has emerged as an important anatomic site for immune surveillance of the CNS (*Alves de Lima et al., 2020*; *Rua and McGavern, 2018*). Soluble brain- and CSF-derived molecules accumulate within the meninges around the dural sinuses and can be captured by local antigen-presenting cells (APCs) (*Louveau et al., 2018b*; *Rustenhoven et al., 2021*). Moreover, in contrast to the brain, which is devoid of lymphatic vessels, the dural meninges develop an extensive network of lymphatic vessels which directly transport soluble brain- and CSF-derived molecules to peripheral lymph nodes (*Aspelund et al., 2015*; *Louveau et al., 2018b*). At homeostasis, a small number of APCs in the dural meninges surround these lymphatic vessels, and upon stimulation these cells can traffic to the deep cervical lymph nodes (*Louveau et al., 2018b*).

Although CD11c$^{hi}$MHC II$^{hi}$ APCs accumulate in the brain during *T. gondii* infection (*John et al., 2011*; *Fischer et al., 2000*), there is only limited evidence to suggest that, in the absence of local lymphatic vasculature, these cells are able to migrate out of the brain to transport antigen directly to peripheral lymph nodes (*Carare et al., 2008*; *Clarkson et al., 2017*). By contrast, professional APCs in the dural meninges are uniquely positioned to sample CNS material and traffic through lymphatic vessels to lymph nodes for T cell activation. Supporting a role for these cells in the immune response to brain infection, we report a striking accumulation of CD11c$^+$ cells in the dural meninges of *CD11c$^{YFP}$* reporter mice chronically infected with the ME49 strain of *T. gondii* (*Figure 1a*). CD11c$^+$ cells localized preferentially around the dural sinuses near meningeal lymphatic vessels (*Figure 1b*), increasing local area coverage from 9% in naïve mice to 44% in chronically infected mice (*Figure 1c*). This accumulation of CD11c$^+$ cells around the sinuses was associated with a threefold increase in the number of CD11c$^+$ cells co-localized to LYVE1$^+$ meningeal lymphatic vessels in infected mice compared to naïve mice (*Figure 1d–e*). This observation suggests that dural APCs access lymphatic vessels in greater numbers in response to brain infection and is consistent with trafficking studies performed previously (*Louveau et al., 2018b*).

To determine whether dendritic cells represent a portion of this emergent population of CD11c$^+$ cells, and to more precisely characterize changes in distinct conventional dendritic cell (cDC) subsets in response to brain infection, we performed spectral flow cytometry on cells purified from the dural meninges of naïve, acutely infected (2 weeks post-infection), or chronically infected (6 weeks post-infection) C57BL/6 mice (*Figure 1—figure supplement 1*). Acute infection represents the period of time, lasting 2–3 weeks, when the parasite spreads systemically throughout the host's peripheral tissues as fast-replicating tachyzoites. Once the parasite is cleared from peripheral tissues by a Th1-driven immune response, chronic infection takes hold and the parasite becomes largely confined to the CNS and skeletal muscle as slow-replicating, cyst-forming bradyzoites (*Saeij et al., 2005*). Our analysis centered on cDC1s, which play a specialized role in the cross-presentation (*Durai and Murphy, 2016*) and are essential for generating early immunity against *T. gondii* (*Poncet et al., 2019*), and cDC2s, which are important activators of CD4 +T cells (*Durai and Murphy, 2016*) but whose role in *T. gondii* infection remains unresolved. While there was no difference in the number of cDC1s (CD45$^+$Lin$^-$CD11c$^{hi}$MHC II$^{hi}$CD64$^-$CD26$^+$XCR1$^+$SIRPα$^-$) or cDC2s (CD45$^+$Lin$^-$CD11c$^{hi}$MHC II$^{hi}$CD64$^-$CD26$^+$XCR1$^-$SIRPα$^+$) in naive and acutely infected mice, by 6 weeks post-infection (wpi) there was a threefold increase in the number of cDC1s and sixfold increase in the number of cDC2s in the dural meninges (*Figure 1f–g*). Moreover, there was a shift in the proportion of cDC1s and cDC2s (*Figure 1—figure supplement 2a-e*). At baseline, cDC2s made up only 20% of the CD26$^+$ cDC population in the dural meninges, but by 6 wpi cDC2s made up 40% of the CD26$^+$ population (*Figure 1—figure supplement 2c*). Intriguingly, the expansion of the dendritic cell compartment in the dural meninges during chronic infection tracked with increasing parasite burden in the brain, which was found to peak at 6 wpi by real-time PCR (*Figure 1h*). By contrast, parasite could not be detected in the dural meninges at 6 wpi by either real-time PCR or immunostaining of PFA-fixed tissue (*Figure 1h*, *Figure 1—figure supplement 2d*). These data suggest that the accumulation of cDC1s and cDC2s in the dural meninges occurred independently of ongoing infection of the dural meninges. This finding reinforces an emerging view that immune cells at border tissues of the CNS, including the dural meninges, play a key role in responding to pathology within the brain parenchyma (*Alves de Lima et al., 2020*; *Rua and McGavern, 2018*; *Rustenhoven et al., 2021*; *Cugurra et al., 2021*; *Pulous et al., 2022*).

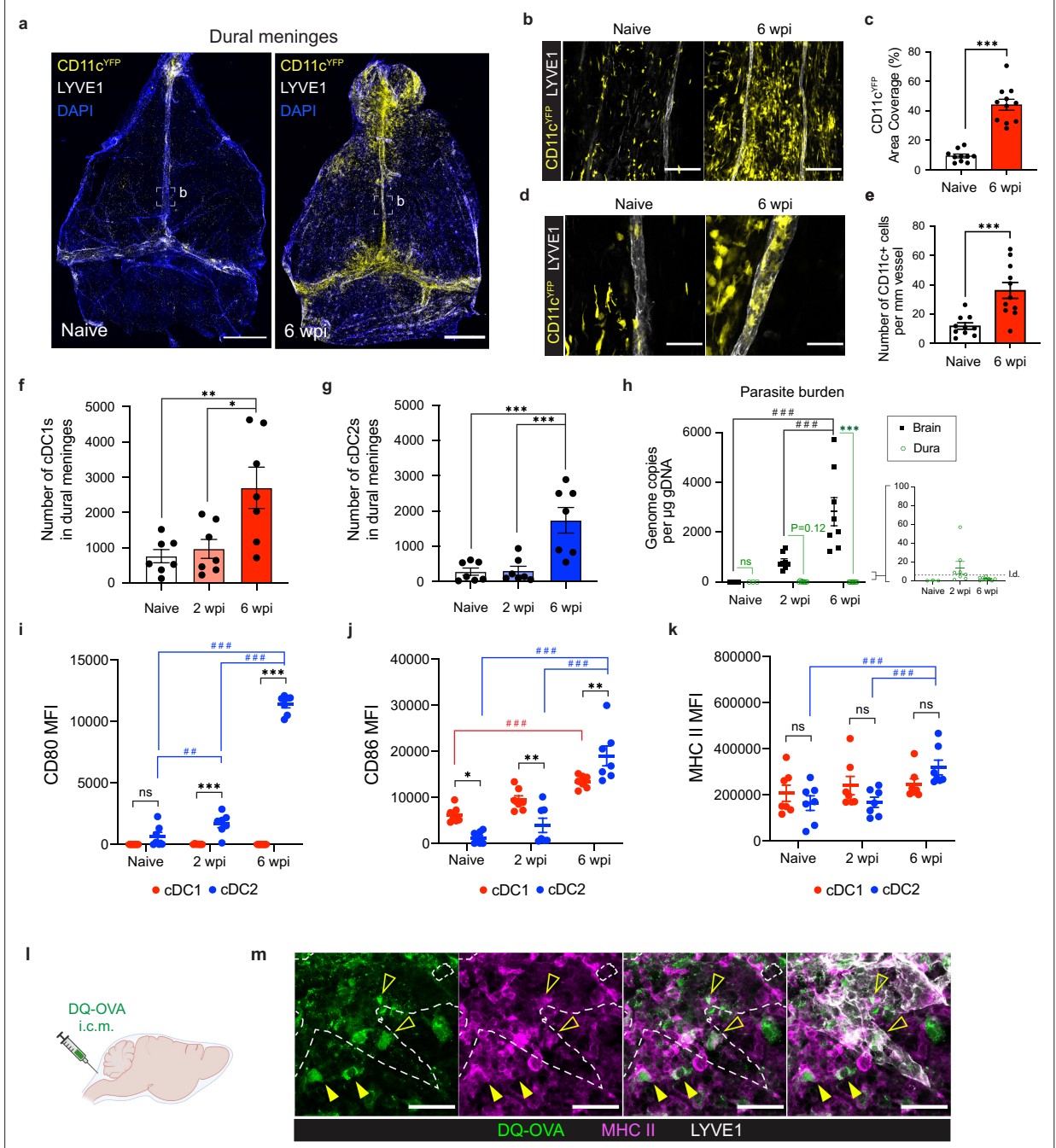

**Figure 1.** Conventional dendritic cells accumulate in the dural meninges during chronic brain infection and sample CSF-derived protein. Mice were infected with 10 cysts of the ME49 strain of *T. gondii* intraperitoneally (i.p.) and analyzed at 6 weeks post-infection. (**a**) Confocal microscopy was used to image whole-mount dural meninges dissected from the skullcaps of naïve or chronically infected *CD11c*$^{YFP}$ reporter mice. Images are representative of three independent experiments. Scale bar, 2000 μm. (**b**) Representative images of CD11c$^+$ cells (yellow) in the region surrounding the dural sinuses. Scale bar, 150 μm. (**c**) Quantification of area coverage by CD11c$^+$ cells in the region surrounding the dural sinuses. Data compiled from three experiments (n = 10-11 mice per group). (**d-e**) Representative images (**d**) and quantification (**e**) of CD11c$^+$ cells (yellow) present within LYVE1$^+$ meningeal lymphatic vessels (white). Data compiled from three experiments (n = 10-11 mice per group). Scale bar, 50 μm. (**f**) Quantification by spectral flow cytometry of total cDC1 number (CD45$^+$Lin$^-$CD11c$^{hi}$MHC II$^{hi}$CD64$^-$CD26$^+$XCR1$^+$SIRPα$^-$) in the dural meninges of naïve, acutely infected, or chronically infected C57BL/6 mice. Data compiled from two experiments (n = 7 mice per group). (**g**) Quantification by spectral flow cytometry of total cDC2 number (CD45$^+$Lin$^-$CD11c$^{hi}$MHC II$^{hi}$CD64$^-$CD26$^+$XCR1$^-$SIRPα$^+$) in the dural meninges of naive, acutely infected, or chronically infected C57BL/6 mice. Data compiled from two experiments (n = 7 mice per group). (**h**) Quantification of *T. gondii* gDNA in brain and dural meninges by real-time PCR. Data compiled from two experiments (n = 3-8 mice per group). (**i–k**) Quantification of the geometric mean fluorescence intensity (MFI) of CD80 (**i**), CD86 (**j**),

*Figure 1 continued on next page*

*Figure 1 continued*

and MHC class II (**k**) expressed by cDC1s and cDC2s in the dural meninges. Data compiled from two experiments (n = 7 mice per group). (**l**) Schematic diagram illustrating intra-cisterna magna (i.c.m.) injection of DQ-OVA into the CSF. (**m**) Representative images of MHC class II-expressing antigen-presenting cells (magenta) co-labeling with fluorescent DQ-OVA cleavage products (green) in the vicinity of (closed arrows) or present within (open arrows) meningeal lymphatic vessels (LYVE1⁺, white). Three independent experiments were performed. Scale bar, 50 µm. Data are represented as mean values ± s.e.m. For **c** and **e**, statistical significance was measured using randomized block ANOVA (two-way), with p<0.001 (***). For (**f and g**) statistical significance was measured using a one-way ANOVA with post-hoc Tukey multiple comparison testing, with p<0.05 (*), p<0.01 (**), and p<0.001 (***). For (**h–k**) statistical significance was measured using a two-way ANOVA, with Tukey's multiple comparison test to assess differences across timepoints [p<0.01 (##) and p<0.001 (###)] and Sidak's multiple comparison test to assess differences between tissues (**h**) or cell type (**i–k**) [ns = not significant, p<0.05 (*), p<0.01 (**), and p<0.001 (***)].

The online version of this article includes the following figure supplement(s) for figure 1:

**Figure supplement 1.** Gating strategy for conventional dendritic cells in the dural meninges.

**Figure supplement 2.** Comparison of cDC1 and cDC2 populations in the dural meninges during chronic brain infection.

**Figure supplement 3.** A population of CD11cʰⁱMHC IIʰⁱ antigen-presenting cells emerges in the cerebrospinal fluid of chronically infected mice.

We next examined the maturation status of dendritic cells (*Hammer and Ma, 2013*) in the dural meninges (*Figure 1i–k*). Consistent with the increase in number of cDC1s and cDC2s at 6 wpi, upregulation of co-stimulatory molecules and MHC class II occurred largely after progression to chronic brain infection (*Figure 1i–k*, *Figure 1—figure supplement 2e-g*). Additionally, cDC2s displayed a much broader degree of activation compared to cDC1s, upregulating expression of CD80, CD86, and MHC class II (*Figure 1i–k*). By contrast, cDC1s expressed minimal levels of CD80 (*Figure 1i*) and showed an increase in CD86 expression but not MHC class II expression (*Figure 1j–k*). These data suggest that chronic brain infection promotes maturation of dendritic cells in the dural meninges but has a greater effect on the activation status of cDC2s than cDC1s.

Because activated dendritic cells introduced into the CSF exit the CNS by accessing meningeal lymphatic vessels (*Louveau et al., 2018b*), it is possible that some of the dendritic cells that we observed in the meninges of infected mice came from the CSF. Therefore, we performed flow cytometry on CSF pooled from 4 to 5 naive or chronically infected mice and saw that, in fact, a population of CD11cʰⁱMHC IIʰⁱ APCs not present in naive mice emerged in the CSF of chronically infected mice (*Figure 1—figure supplement 3a-b*), expressing high levels of CD80 and CD86 (*Figure 1—figure supplement 3c-e*). Interestingly, the concentration of CSF-borne APCs was consistent with that reported by a study of human CSF samples isolated from patients with Lyme neuroborreliosis (*Pashenkov et al., 2001*).

We next sought to determine whether APCs in the dural meninges are able to transport CSF-derived protein during infection. CSF has been shown to function as a sink for brain-derived macromolecules (*Abbott, 2004*; *Plog and Nedergaard, 2018*) and capturing CSF-borne protein represents a potential mechanism by which APCs in the dural meninges could sample brain-derived antigen and transport it to lymph nodes in the periphery. To address this question, soluble DQ-OVA was injected into the CSF of chronically infected mice via intra-cisterna magna (i.c.m.) injection (*Figure 1l*). DQ-OVA is a self-quenched conjugate of ovalbumin labeled with BODIPY dyes that only emits fluorescence after proteolytic cleavage. DQ-OVA has been commonly used to examine uptake and processing of soluble antigen by dendritic cells (*Ling et al., 2003*; *Gerner et al., 2017*; *Sixt et al., 2005*). Within 2 hr of i.c.m. injection, we found that 30% of cDC1s and cDC2s in the dural meninges harbored DQ-OVA cleavage products (*Figure 1—figure supplement 2h-j*). After 24 hr, cleavage products were detected in ~5% of cDC1s and cDC2s (*Figure 1—figure supplement 2h-j*), which may reflect egress of DQ-OVA +dendritic cells from the meninges or loss of the BODIPY fluorescent signal over time. Imaging of whole mount dural meninges revealed that DQ-OVA⁺ cells were distributed along and within lymphatic vessels after 12 hr, suggesting that dural APCs capture and transport CSF-borne antigen during chronic brain infection (*Figure 1m*). All together, these findings highlight significant changes in the APC compartment of the dural meninges in response to *T. gondii* brain infection. Type 1 and type 2 conventional dendritic cells increased in number in the dural meninges, displayed a more activated phenotype, and were able to sample CSF-borne protein in the vicinity of meningeal lymphatic vessels.

## T cell responses in the deep cervical lymph nodes peak following progression to chronic brain infection

Initial studies in mice, and later in humans, have demonstrated that meningeal lymphatic vessels drain directly to the deep cervical lymph nodes (DCLN) (*Albayram et al., 2022*; *Louveau et al., 2018b*). Even before the immunologic function of the meningeal lymphatic system started coming into focus, these lymph nodes were strongly implicated in regulating CNS immunity. For example, injection of antigen into rodent brains elicited strong humoral responses in the DCLNs (*Harling-Berg et al., 1989*), and surgical excision of the DCLNs contributed to a reduction in severity of experimental autoimmune encephalomyelitis (*van Zwam et al., 2009*; *Furtado, 2008*). To date, few studies have examined the contribution of the deep cervical lymph nodes to immunity against CNS pathogens. To address the role of the DCLNs during *T. gondii* infection, we first confirmed that during chronic infection CSF-derived protein could be sampled by APCs within these lymph nodes. Upon i.c.m. injection of DQ-OVA into chronically infected mice, 3% of CD11c$^{hi}$MHC II$^{hi}$ APCs in the DCLNs were found to harbor digested protein products after 5 hr (*Figure 2a–b*). By contrast, processed DQ-OVA was not detected in the inguinal lymph nodes (ILN), which drain peripheral tissue (*Figure 2a–b*). Further characterization of the DQ-OVA$^+$ APC populations in the DCLNs revealed that equal proportions of cDC1s and cDC2s harbored digested protein products both at 2 hr and 24 hr after i.c.m. injection (*Figure 2—figure supplement 1a-c*), the latter time point representing when migratory dendritic cells trafficking from the meninges have been shown to reach the DCLNs (*Louveau et al., 2018b*).

Based on these results, we next sought to understand the kinetics of T cell activation in the deep cervical lymph nodes. For comparison, we also examined the kinetics of T cell activation in the inguinal lymph nodes. When C57BL/6 mice were infected with the ME49 strain of *T. gondii*, activated (CD44$^{hi}$CD62L$^{lo}$) CD4$^+$ and CD8$^+$ T cell number in the ILNs was greatest during acute infection (2 wpi) (*Figure 2c–d*, *Figure 2—figure supplement 2a*), consistent with the broad distribution of parasite in peripheral tissues at this time point. Conversely, only a small number of activated CD4$^+$ and CD8$^+$ T cells was detected in the DCLNs at 2 wpi (*Figure 2c–d*). Expansion of CD4$^+$ and CD8$^+$ T cells in the DCLNs became more pronounced after progression to the chronic stage of infection, with the peak in number of activated CD4$^+$ and CD8$^+$ T cells occurring at 6–8 wpi (*Figure 2c–d*). To better understand the stage-dependent activation of T cells in the DCLNs, we used MHC class I tetramer to track a parasite epitope (SIINFEKL)-specific population of CD8$^+$ T cells generated in response to infection with Pru-OVA, a recombinant type II strain of *T. gondii* engineered to express a secreted, truncated form of the model antigen ovalbumin (*Pepper et al., 2004*; *Figure 2—figure supplement 2b*). Consistent with the polyclonal CD8$^+$ T cell response observed in response to ME49 infection (*Figure 2d*), the peak SIINFKEKL-specific CD8$^+$ T cell response in the ILNs occurred during acute infection (2–3 wpi), when very few parasite-specific T cells were detectable in the DCLNs (*Figure 2e*). By contrast, expansion of parasite-specific CD8$^+$ T cells in the DCLNs was greatest during chronic infection, increasing significantly between 3 and 6 wpi (*Figure 2e*). These results likely reflect the distinct sources of antigen that each lymph node site drains during the different stages of infection. In support of this, we found that parasite-specific T cell responses in the DCLNs tracked with parasite burden in the brain (*Figure 2f*), whereas parasite-specific T cell responses in the ILNs tracked with parasite burden in the ILNs. During acute infection, ILNs appear to drain parasite from infected peripheral tissues or become directly infected themselves (*Figure 2f*). Notably, the DCLNs did not show high levels of parasite burden during acute infection, suggesting that hematogenous routes of dissemination did not provide significant access of the parasite to the DCLNs.

To understand how the quality of T cell responses generated in the DCLNs and brain change as infection progresses from the acute to chronic stage, we assessed expression of co-inhibitory molecules, which can become upregulated in the setting of persistent infection (*Attanasio and Wherry, 2016*), and markers that define the effector and memory T cell subsets during *T. gondii* infection (*Chu et al., 2016*; *Landrith et al., 2017*; *Figure 2—figure supplement 2c*). We observed minimal expression of co-inhibitory molecules by endogenous and transferred SIINFEKL-specific CD8 + cells (*Figure 2—figure supplement 3*), and as expected we observed an increase in the proportion of memory-like T cell subsets during chronic infection (*Figure 2—figure supplement 4*).

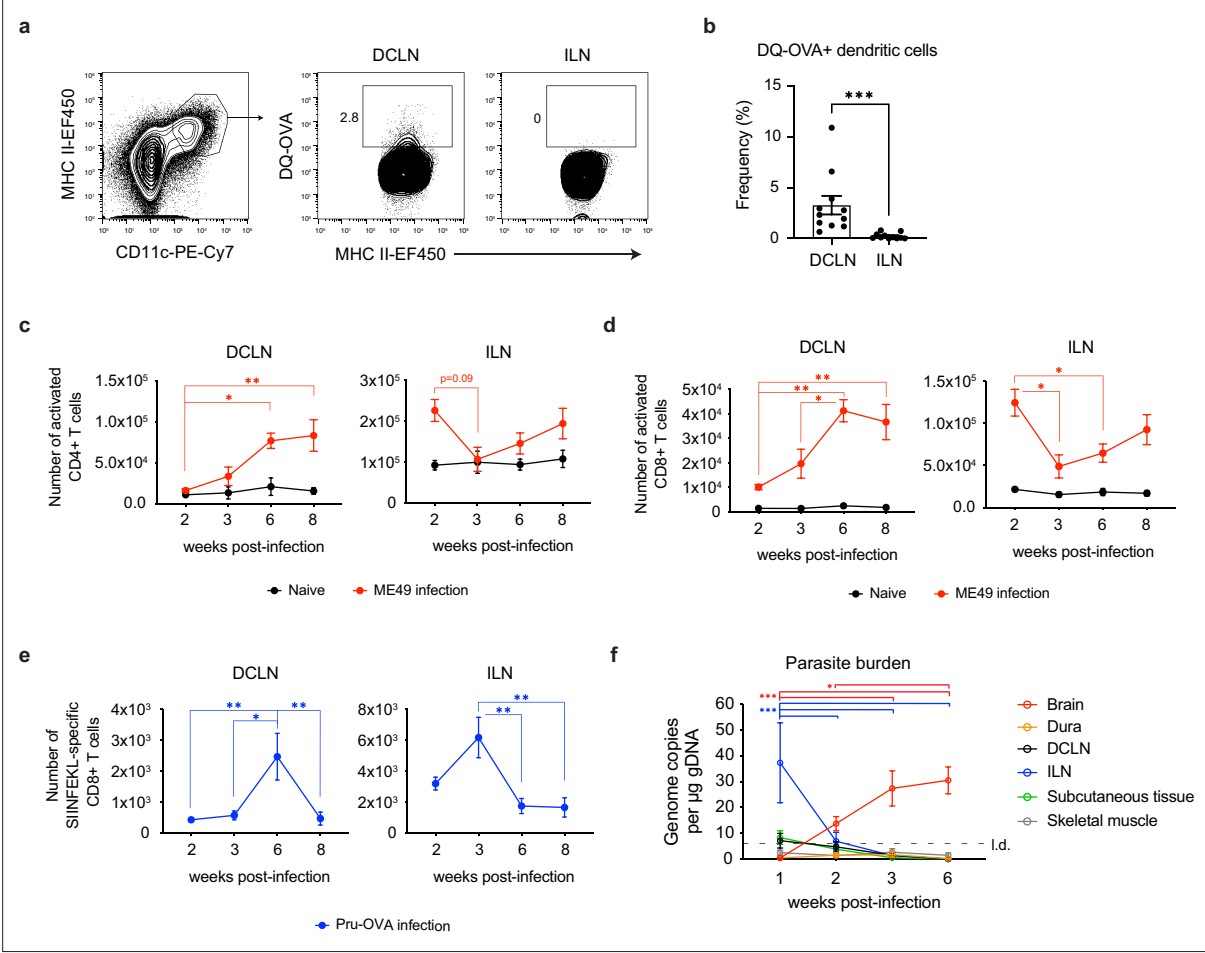

**Figure 2.** Expansion of T cells in the deep cervical lymph nodes occurs primarily during the chronic stage of infection and tracks with parasite burden in the brain. (**a-b**) DQ-OVA was injected into the CSF of chronically infected mice by i.c.m. injection and fluorescent emission of proteolytically cleaved DQ-OVA was measured in CD11c(hi)MHC II(hi) antigen-presenting cells of the deep cervical lymph nodes (DCLNs) and inguinal lymph nodes (ILNs) by flow cytometry. (**a**) Representative contour plots of DQ-OVA+ antigen-presenting cells in the DCLNs or ILNs at 6 wpi. CD11c(hi)MHC II(hi) cells were pre-gated on singlets/live/TCRβ⁻/NK1.1⁻/CD19⁻. (**b**) Quantification of frequency of DQ-OVA+ antigen-presenting cells in the DCLNs or ILNs at 6 wpi. Data are compiled from three experiments (n = 11 mice per group) and are represented as mean values ± s.e.m. Statistical significance was measured using randomized block ANOVA (two-way), with p<0.001 (***). (**c-d**) C57BL/6 mice were infected i.p. with 10 cysts of the ME49 strain of *T. gondii* and total number of activated CD4+ T cells (**c**) or activated CD8+ T cells (**d**) in the DCLNs or ILNs was quantified at multiple time points over the course of acute and chronic infection (red dots). The steady-state number of activated CD4+ and CD8+ T cells in the different lymph node compartments was measured in naïve mice at corresponding time points (black dots). Activated T cells displayed a CD44(hi)CD62L(lo) phenotype. Data are compiled from three experiments and are represented as mean values ± s.e.m. (n = 6-14 mice per group per timepoint). (**e-f**) C57BL/6 mice were infected i.p. with 1,000 tachyzoites of the Pru-OVA strain of *T. gondii*. (**e**) Total number of SIINFEKL-specific CD8+ T cells in the DCLNs or ILNs was quantified at multiple time points over the course of acute and chronic infection using tetramer reagent. Data are compiled from two experiments and are represented as mean values ± s.e.m. (n = 7 mice per timepoint). (**f**) Quantification of *T. gondii* gDNA in brain (red), dural meninges (orange), deep cervical lymph nodes (black), inguinal lymph nodes (blue), subcutaneous adipose tissue isolated from the flank (green), and quadriceps femoris skeletal muscle tissue (gray) by real-time PCR. Data are compiled from two experiments and are represented as mean values ± s.e.m. (n = 5-6 mice per tissue per timepoint). For (**c-e**), statistical significance of differences across time points in infected mice was measured using one-way ANOVA with post-hoc Tukey multiple comparison testing. p<0.05 (*) and p<0.01 (**). For (**f**) statistical analysis was performed using a two-way ANOVA with Tukey's multiple comparison test to assess differences across timepoints. Statistically significant differences are indicated, with p<0.05 (*) and p<0.001 (***).

The online version of this article includes the following source data and figure supplement(s) for figure 2:

**Source data 1.** CD4+ T cell activation in the DCLNs and ILNs over the course *T. gondii* infection.

**Source data 2.** CD8+ T cell activation in the DCLNs and ILNs over the course of *T. gondii* infection.

**Source data 3.** SIINFEKL-specific CD8+ T cell responses in the DCLNs and ILNs over the course of *T. gondii* infection.

**Source data 4.** Parasite burden in the brain and peripheral tissues over the course of *T. gondii* infection.

*Figure 2 continued on next page*

Figure 2 continued

**Figure supplement 1.** Uptake of cerebrospinal fluid-borne protein by cDC1s and cDC2s of the deep cervical lymph nodes during chronic brain infection.

**Figure supplement 2.** Gating strategy for CD4+ and CD8+ T cell responses following infection with ME49 or Pru-OVA.

**Figure supplement 3.** Co-inhibitory molecule expression of endogenous and transferred SIINFEKL-specific CD8+ T cells during acute and chronic infection.

**Figure supplement 4.** Effector or memory-like status of endogenous and transferred SIINFEKL-specific CD8+ T cells during acute and chronic infection.

## Meningeal lymphatic drainage promotes peripheral T cell responses against *T. gondii*

To directly test the function of meningeal lymphatic drainage during *T. gondii* brain infection, we surgically ligated the collecting vessels afferent to the deep cervical lymph nodes. This approach has been used to test the function of meningeal lymphatic drainage in experimental models of multiple sclerosis, glioblastoma, and acute viral infection of the brain (*Song et al., 2020*; *Louveau et al., 2018b*; *Li et al., 2022*). We performed ligation at 3 wpi, when T cells begin to display a response against the parasite in the DCLNs (*Figure 2*), and analyzed mice 3 weeks later (*Figure 3a*). Unless otherwise stated, experiments were performed using the ME49 strain of *T. gondii*. Tracer studies using Evans blue or Alexa Fluor 594-conjugated ovalbumin (OVA-AF594) confirmed the efficacy of the procedure and demonstrated a greater than 95% reduction in outflow of CSF-derived components to the DCLNs in ligated animals (*Figure 3—figure supplement 1a-b*). Importantly, ligation did not affect outflow of CSF to the superficial cervical lymph nodes, another drainage site for CSF-borne protein (*Figure 3—figure supplement 1c*; *Ma et al., 2017*).

Given the pronounced accumulation of cDC1s and cDC2s in the dural meninges during chronic brain infection (*Figure 1f–g*), we first examined the effect of meningeal lymphatic drainage on dendritic cell responses in the deep cervical lymph nodes. Interestingly, restriction of meningeal lymphatic drainage by ligation had no effect on the total number of cDC1s and cDC2s in the DCLNs (*Figure 3b–c*) but did result in a reduction in the frequency and number of CD103+ cDC1 s and CD103+ cDC2s at this site (*Figure 3d–i*), consistent with a migratory function for this subset of dendritic cells (*Merad et al., 2013*; *Mildner and Jung, 2014*). Additionally, even though there was no difference in the overall number of cDC1s and cDC2s in the DCLNs, the expression of co-stimulatory molecules CD80 and CD86 by both cDC1s and cDC2s was significantly reduced after ligation (*Figure 3j–q*). These data suggest that in the setting of brain infection meningeal lymphatic drainage promotes maturation of dendritic cells at this site.

We next hypothesized that peripheral T cell responses would be significantly impaired by ligation, given the defects observed in the dendritic cell populations. Indeed, restricting meningeal lymphatic outflow led to a significant decrease in the frequency and number of CD4+ and CD8+ T cells in the DCLNs expressing the early activation marker CD69, despite equal numbers of CD4+ and CD8+ T cells being present in the lymph nodes after ligation (*Figure 4a–b*, *Figure 4—figure supplement 1a-f*). Similarly, when Pru-OVA was used to infect mice, ligation caused a decrease in expression of CD69 by SIINFEKL-specific CD8+ T cells in the DCLNs (*Figure 4c–d*). To assess whether this decrease in activation was associated with reduced T cell proliferation, we measured Ki67 expression by flow cytometry and observed a concomitant decrease in the frequency of proliferative CD4+ and CD8+ T cells in ligated mice (*Figure 4e–h*). We also examined expression of co-inhibitory molecules on CD4+ and CD8+ T cells in the DCLNs and saw a reduction in expression of PD-1, LAG-3, and TIM-3 after ligation (*Figure 4—figure supplement 2*). Of note, T cell activation in the superficial cervical lymph nodes, which drain facial tissues and serve as an additional drainage site for CSF (*Ma et al., 2017*; *Lohrberg and Wilting, 2016*), was not affected (*Figure 4—figure supplement 1g-i*).

During *T. gondii* infection, CD4+ and CD8+ T cells confer protection to the host by producing large amounts of IFN-γ and TNF-α, cytokines that activate intracellular mechanisms of parasite restriction in infected host cells (*Schlüter et al., 2003*; *Wang et al., 2004*). To determine whether meningeal lymphatic drainage is required for maintaining high-quality peripheral CD4+ and CD8+ T cell responses, intracellular cytokine staining was performed on cells that had been isolated from the deep cervical lymph nodes of sham-operated or ligated mice and restimulated ex vivo for 24 hr with soluble tachyzoite antigen (STAg). Remarkably, IFN-γ production was reduced by 69% among CD4+ T

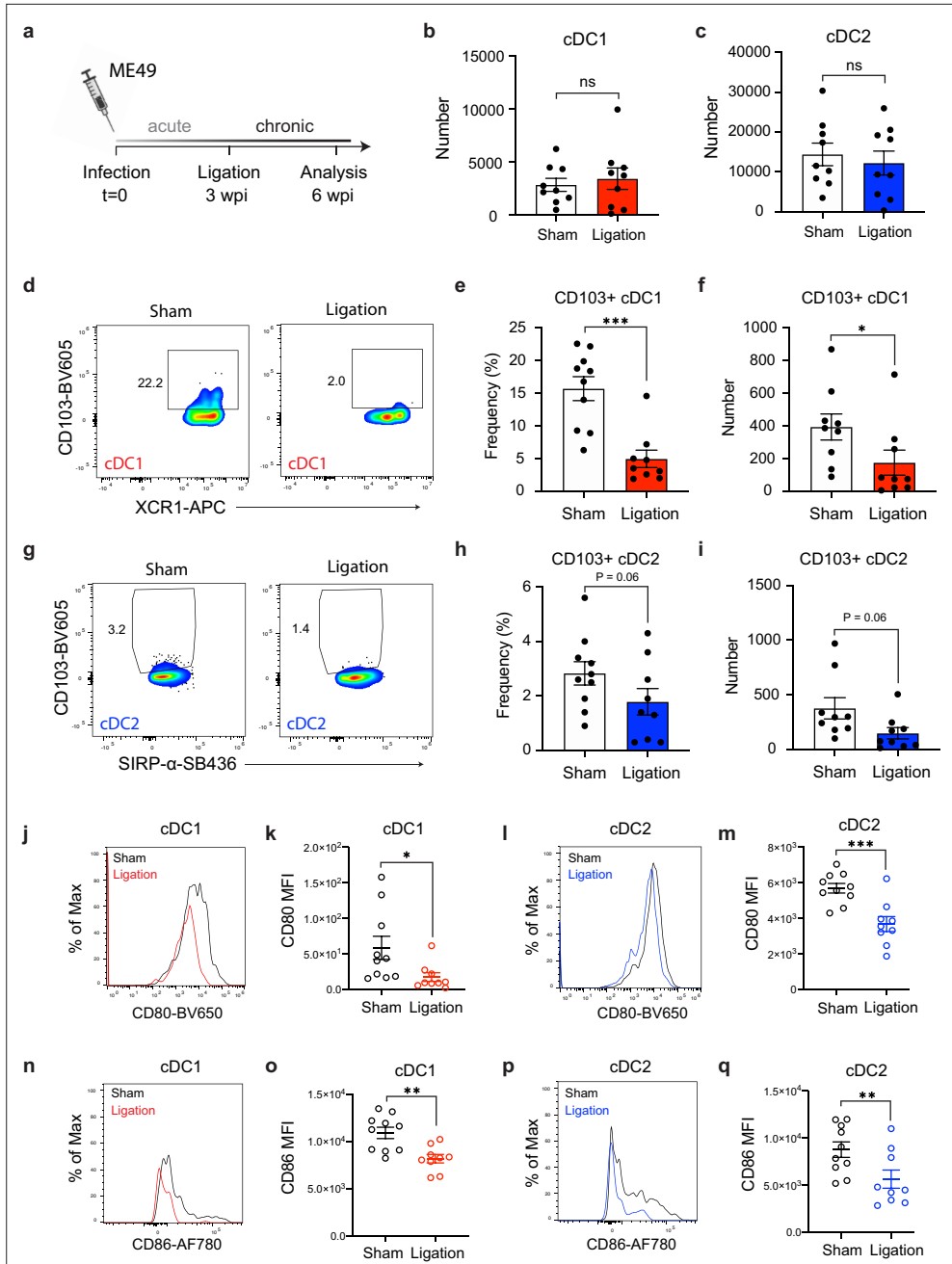

**Figure 3.** Restricting meningeal lymphatic drainage disrupts dendritic cell activation in the deep cervical lymph nodes. Chronically infected C57BL/6 mice were subjected to surgical ligation of collecting vessels afferent to the DCLNs or sham surgery. (**a**) Experimental design for ligation studies in C57BL/6 mice. (**b–c**) Quantification by spectral flow cytometry of total cDC1 (**b**) and cDC2 (**c**) number observed in the DCLNs three weeks after ligation or sham surgery. Data are compiled from two experiments (n = 9 mice per group). (**d-f**) Representative dot plots (**d**) showing frequency of CD103+ cDC1s in the DCLNs three weeks after ligation or sham surgery, with quantification of frequency (**e**) and number (**f**) compiled from two independent experiments (n = 9-10 mice per group). (**g–i**) Representative dot plots (**g**) showing frequency of CD103+ cDC2s in the DCLNs three weeks after ligation or sham surgery, with quantification of frequency (**h**) and number (**i**) compiled from two independent experiments (n = 9-10 mice per group). (**j–k**), Representative flow histograms showing expression of CD80 by cDC1s in the DCLNs 3 weeks after ligation or sham surgery (**j**) and quantification of the geometric mean fluorescence intensity (**k**). Data compiled from two experiments (n = 9-10 mice per group). l-m, Representative flow histograms showing expression of CD80 by cDC2s in the DCLNs 3 weeks after ligation or sham surgery (**l**) and quantification of the geometric mean fluorescence intensity (**m**). Data compiled from two experiments (n = 9-10 mice per group).

*Figure 3 continued on next page*

*Figure 3 continued*

(**n–o**) Representative flow histograms showing expression of CD86 by cDC1s in the DCLNs 3 weeks after ligation or sham surgery (**n**) and quantification of the geometric mean fluorescence intensity (**o**). Data compiled from two experiments (n = 9-10 mice per group). (**p–q**) Representative flow histograms showing expression of CD86 by cDC2s in the DCLNs 3 weeks after ligation or sham surgery (**p**) and quantification of the geometric mean fluorescence intensity (**q**). Data compiled from two experiments (n = 9-10 mice per group). Data are represented as mean values ± s.e.m. and statistical significance was measured using randomized block ANOVA (two-way), with ns = not significant, p<0.05 (*), p<0.01 (**), and p<0.001 (***).

The online version of this article includes the following figure supplement(s) for figure 3:

**Figure supplement 1.** Tracer studies confirm disruption of meningeal lymphatic drainage to the deep cervical lymph nodes by ligation surgery.

cells and 60% among CD8$^+$ T cells after ligation surgery (*Figure 4i–n*). TNF-α production was similarly impaired in CD4$^+$ T cells isolated from the DCLNs of ligated mice (*Figure 4o–q*). IFN-γ and TNF-α production was not detected in unstimulated CD4$^+$ and CD8$^+$ T cells, suggesting that T cell responses were parasite-specific (*Figure 4j, m and p*).

All together, we can conclude that meningeal lymphatic drainage contributes to robust parasite-specific CD4$^+$ and CD8$^+$ T cell responses in the deep cervical lymph nodes during chronic brain infection. Restriction of meningeal lymphatic outflow caused defects in dendritic cell responses and peripheral CD4$^+$ and CD8$^+$ T cell activation, proliferation, and cytokine production.

## Meningeal lymphatic drainage is dispensable for host protection of the brain

Several recent studies have demonstrated that meningeal lymphatic drainage plays a key role in supporting T cell responses in the brain against locally injected tumor cells (*Song et al., 2020*; *Hu et al., 2020*). Moreover, it has been shown that meningeal lymphatic drainage promotes host survival during acute brain infection with Japanese encephalitis virus (*Li et al., 2022*). Based on these studies, we hypothesized that meningeal lymphatic drainage would be required for host-protective T cell responses in the brain against *T. gondii*. Unexpectedly, restricting meningeal lymphatic drainage had no discernible impact on the magnitude or quality of CD4$^+$ and CD8$^+$ T cell responses in the brain after ligation, despite impairing T cell responses in the deep cervical lymph nodes (*Figure 5a–e*).

Indeed, the total number of CD4$^+$ and CD8$^+$ T cells in the brain remained unchanged between sham and ligated mice (*Figure 5a–b*), and the frequency of activated (CD44$^{hi}$CD62L$^{lo}$) cells did not change (*Figure 5—figure supplement 1a-d*). To determine whether there were differences in the quality of T cell responses in the brain, we measured the total number of IFN-γ-producing CD4$^+$ and CD8$^+$ T cells in the brain following ex vivo restimulation with STAg, but again saw no change after ligation (*Figure 5c–d*). We also measured the expression of co-inhibitory molecules and memory markers on CD4$^+$ and CD8$^+$ T cells and observed no differences (*Figure 5—figure supplement 2*, *Figure 5—figure supplement 3*). Finally, no differences were detected in the total number of SIINFEKL-specific CD8$^+$ T cells in the brains of sham-operated or ligated mice chronically infected with Pru-OVA (*Figure 5e*). Because parasite burden was similar between the two groups (*Figure 5f*, *Figure 5—figure supplement 1e*), it is unlikely that any other mechanism of parasite control became defective in the brain as a consequence of ligation.

Since CSF-borne antigen can drain to the superficial cervical lymph nodes (SCLN) (*Figure 3—figure supplement 1c*; *Ma et al., 2017*) in addition to the DCLNs, we next sought to determine whether the SCLNs could be compensating for any loss in T cell activation that occurs in the DCLNs. We therefore performed surgical ligation of the afferent lymphatic vessels that carry CSF to both the DCLNs and SCLNs in order to provide more comprehensive restriction of CSF outflow (*Figure 5—figure supplement 4a-b*). However, even with ligation of vessels draining to both groups of lymph nodes, there was no discernible difference in the magnitude of the CD4$^+$ or CD8$^+$ T cell response in the brain (*Figure 5—figure supplement 4c-d*).

These data indicate that meningeal lymphatic drainage is not necessary for maintenance of the host-protective T cell population in the brain during chronic infection with *T. gondii*. These results contrast with recent studies showing that T cell responses against brain tumors and host protection

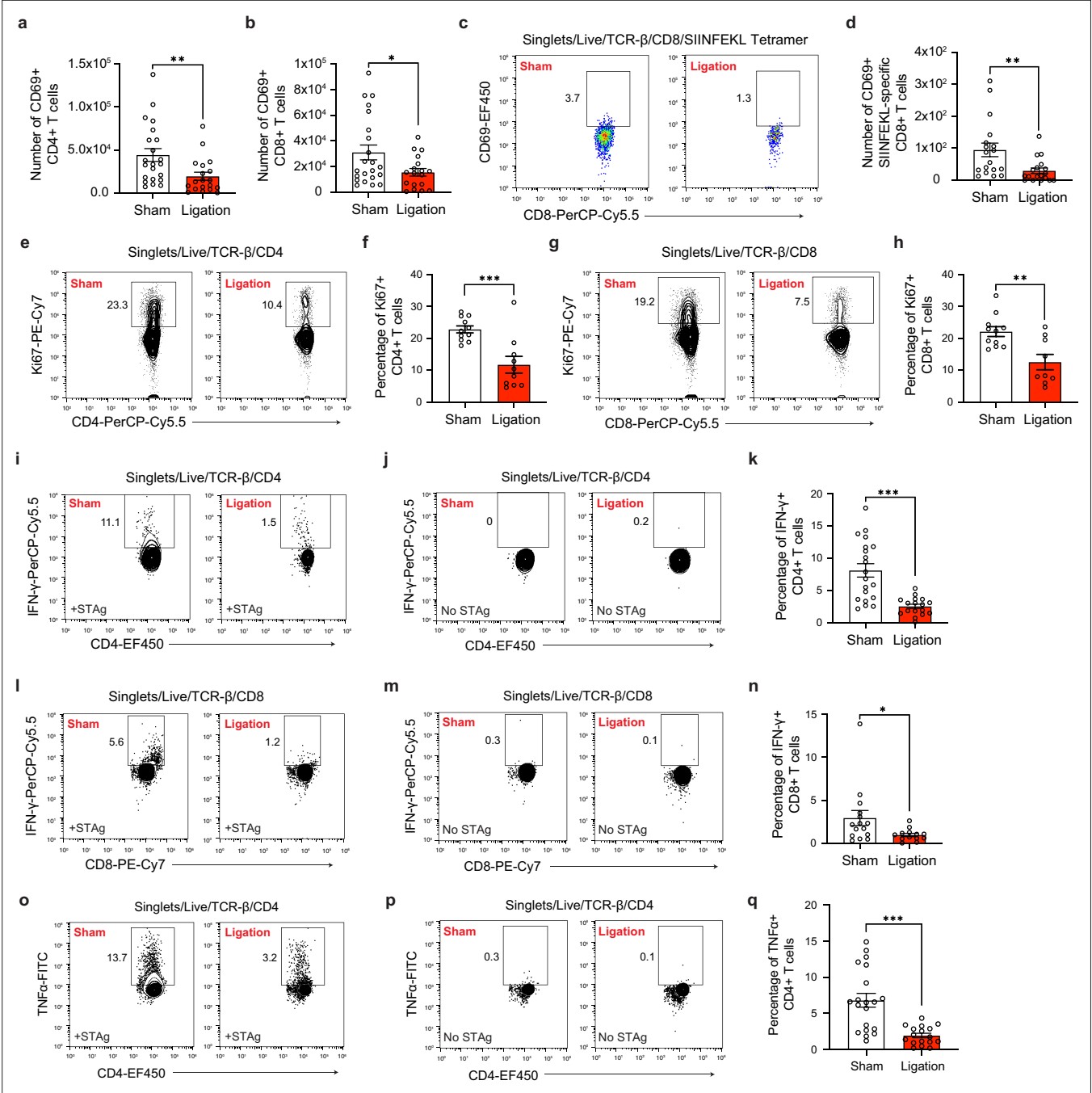

**Figure 4.** Restricting meningeal lymphatic drainage disrupts T cell activation, proliferation, and cytokine production in the deep cervical lymph nodes. Chronically infected C57BL/6 mice were subjected to surgical ligation of collecting vessels afferent to the DCLNs or sham surgery. (**a–b**) Quantification by flow cytometry of the number of CD69-expressing CD4+ T cells (**a**) or CD8+ T cells (**b**) in the DCLNs 3 weeks after ligation or sham surgery. Data are compiled from four experiments (n = 18-22 mice per group). (**c–d**) Representative dot plots (**c**) and total number (**d**) of CD69-expressing SIINFEKL-specific CD8+ T cells in the DCLNs 3 weeks after ligation or sham surgery of mice infected with Pru-OVA. Data are compiled from four experiments (n = 18 mice per group). (**e–f**) Representative contour plots (**e**) and average frequency (**f**) of Ki67-expressing CD4+ T cells in the DCLNs 3 weeks after ligation or sham surgery. Data are compiled from two experiments (n = 10-11 mice per group). (**g–h**) Representative contour plots (**g**) and average frequency (**h**) of Ki67-expressing CD8+ T cells in the DCLNs 3 weeks after ligation or sham surgery. Data are compiled from two experiments (n = 9-11 mice per group). (**i–q**) Intracellular staining of IFN-γ and TNF-α was performed on cells isolated from the DCLNs of ligated or sham-operated mice 24 hr after ex vivo restimulation with soluble tachyzoite antigen (STAg) or media alone (no STAg). Representative contour plots show expression of IFN-γ by CD4+ T cells following STAg restimulation (**i**) or in the presence of media alone (**j**), with quantification of the frequency of IFN-γ+ CD4+ T cells following STAg restimulation (**k**) (four experiments, n = 17-20 mice per group). Representative contour plots show expression of IFN-γ by CD8+ T cells following STAg

*Figure 4 continued on next page*

*Figure 4 continued*

restimulation (**l**) or in the presence of media alone (**m**), with quantification of the frequency of IFN-γ⁺ CD8⁺ T cells following STAg restimulation (**n**) (three experiments, n = 13-15 mice per group). Representative contour plots show expression of TNF-α by CD4⁺ T cells following STAg restimulation (**o**) or in the presence of media alone (**p**), with quantification of the frequency of TNF-α⁺ CD4⁺ T cells following STAg restimulation (**q**) (four experiments, n = 17-20 mice per group). Statistical significance was measured using randomized block ANOVA (two-way), with p<0.05 (\*), p<0.01 (\*\*), and p<0.001 (\*\*\*). Data are represented as mean values ± s.e.m.

The online version of this article includes the following figure supplement(s) for figure 4:

**Figure supplement 1.** Restricting meningeal lymphatic drainage impairs T cell activation in the deep cervical lymph nodes without affecting T cell activation in the superficial cervical lymph nodes.

**Figure supplement 2.** Restricting meningeal lymphatic drainage disrupts co-inhibitory molecule upregulation on T cells in the deep cervical lymph nodes.

against acute viral infection of the brain are supported by meningeal lymphatic drainage to the deep cervical lymph nodes (*Song et al., 2020*; *Li et al., 2022*; *Hu et al., 2020*).

## Antigen-dependent stimulation of T cells occurs in CNS- and non-CNS-draining lymph nodes during chronic infection

Because T cell recruitment from the periphery has been demonstrated to be indispensable for T cell responses against *T. gondii* in the brain (*Harris et al., 2012*; *Wilson et al., 2009*), it is likely that T

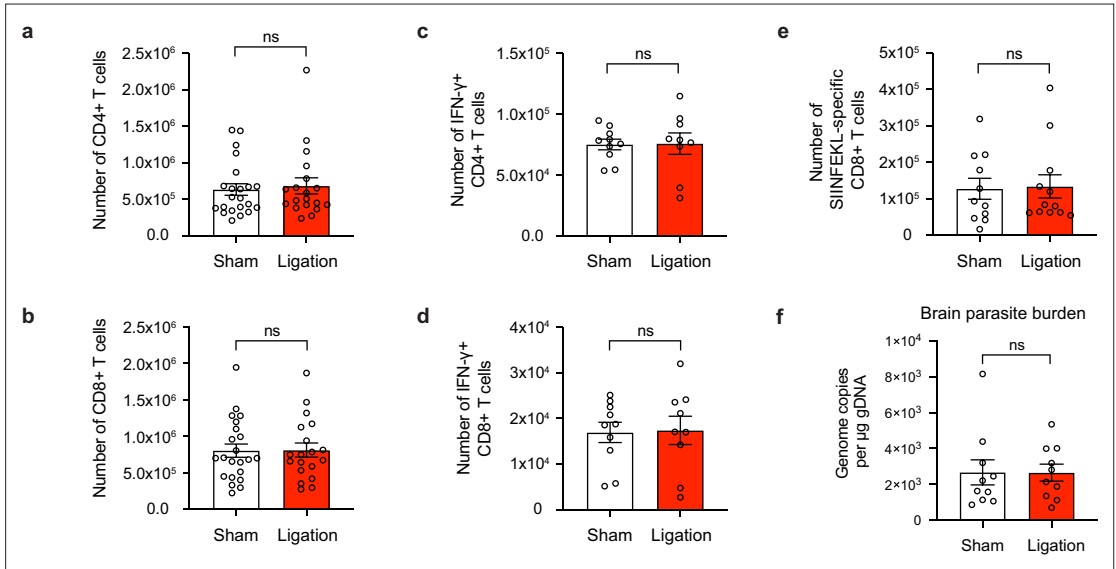

**Figure 5.** Meningeal lymphatic drainage is dispensable for host-protective T cell responses in the brain. Chronically infected C57BL/6 mice were subjected to surgical ligation of collecting vessels afferent to the DCLNs or sham surgery. (**a–b**), Quantification by flow cytometry of the number of infiltrating CD4⁺ T cells (**a**) and CD8⁺ T cells (**b**) in the brain tissue of ligated or sham-operated mice. Data are compiled from four experiments (n = 19-22 mice per group). (**c–d**), Intracellular staining of IFN-γ was performed on mononuclear cells isolated from brains of ligated or sham-operated mice 24 hr after ex vivo restimulation with soluble tachyzoite antigen (STAg). Total number of CD4⁺ T cells (**c**) and CD8⁺ T cells (**d**) expressing IFN-γ after STAg restimulation are shown. Data are compiled from two experiments (n = 9-10 mice per group). (**e**), Quantification by flow cytometry of the number of SIINFEKL-specific CD8⁺ T cells in the brain tissue of ligated or sham-operated mice chronically infected with Pru-OVA. Data are compiled from three experiments (n = 11-12 mice per group). (**f**), Quantification of *T. gondii* gDNA in the brain tissue of ligated or sham-operated mice by real-time PCR. Data are compiled from two experiments (n = 10 mice per group). Statistical significance was measured using randomized block ANOVA (two-way), with ns = not significant. Data are represented as mean values ± s.e.m.

The online version of this article includes the following figure supplement(s) for figure 5:

**Figure supplement 1.** T cell activation and cyst burden are unaffected in the brain after ligation.

**Figure supplement 2.** T cells express similar levels of co-inhibitory molecules in the brain after ligation.

**Figure supplement 3.** Ligation does not affect effector or memory-like status of T cells in the brain.

**Figure supplement 4.** Restricting meningeal lymphatic drainage to the DCLNs and SCLNs does not affect the magnitude of the T cell response in the brain.

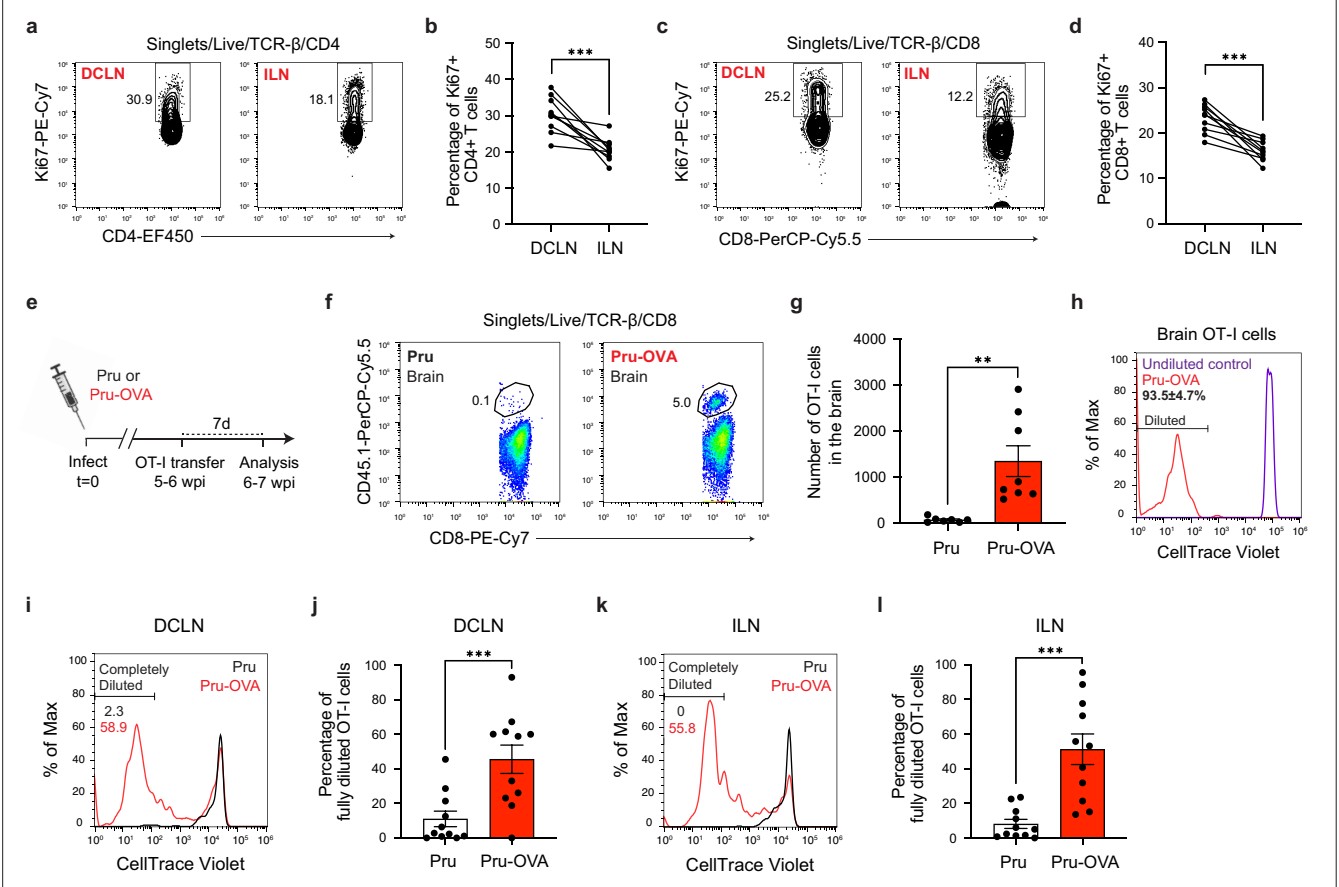

**Figure 6.** Antigen-dependent proliferation of T cells occurs in CNS- and non-CNS-draining lymph nodes during chronic infection with *T. gondii*. T cell proliferation was measured at 6 wpi in the deep cervical lymph nodes (DCLNs) and a representative group of non-CNS-draining lymph nodes, the inguinal lymph nodes (ILNs). (a–d), Representative contour plots (a, c) and average frequency (b, d) of Ki67-expressing CD4+ or CD8+ T cells in the different lymph node compartments of C57BL/6 mice chronically infected with the ME49 strain of *T. gondii*. Data are compiled from two experiments (n = 10 mice per group) and statistical significance was measured using a two-tailed paired *t*-test, with p<0.001 (***). (e–l), C57BL/6 mice were infected with the OVA-secreting Pru strain of *T. gondii* (Pru-OVA) or the control parental Pru (Pru) strain of *T. gondii*. 5 to 6 weeks later, OT-I cells (CD45.1 congenic) labeled with CellTrace Violet were transferred intravenously, and tissues were analyzed 7 days post-transfer. (e), Experimental design for OT-I transfer studies. (f–g), Representative dot plots (f) and total number (g) of OT-I cells infiltrating the brain tissue of mice infected with Pru-OVA or parental Pru. Data are compiled from two experiments (n = 7-8 mice per group). (h), Representative flow histogram showing CellTrace Violet dye dilution among brain-infiltrating OT-I cells of mice chronically infected with Pru-OVA or parental Pru. The average frequency (mean value ± s.e.m.) of fully diluted cells was calculated from two pooled experiments (n = 8 mice). (i–j), Representative flow histogram (i) and average frequency (j) of fully diluted OT-I cells in the DCLNs of mice chronically infected with Pru-OVA or parental Pru. Data are compiled from three experiments (n = 11 mice per group). (k–l), Representative flow histogram (k) and average frequency (l) of fully diluted OT-I cells in the ILNs of mice chronically infected with Pru-OVA or parental Pru. Data are compiled from three experiments (n = 11 mice per group). For all experiments data are represented as mean values ± s.e.m. and for g, j, and l statistical significance was measured using randomized block ANOVA (two-way), with p<0.01 (**) and p<0.001 (***).

The online version of this article includes the following figure supplement(s) for figure 6:

**Figure supplement 1.** Assessing Nur77 reporter expression in mice chronically infected with Pru-OVA or parental Pru.

cell activation during the chronic stage of infection is not limited to the CNS-draining lymph nodes. In order to identify alternative sources of T cells in the periphery, we began by performing a pairwise comparison of T cell proliferation in the deep cervical lymph nodes and inguinal lymph nodes of ME49-infected mice at 6 wpi. A large percentage of CD4+ and CD8+ T cells expressed Ki67 in the DCLNs at this time point, indicating robust T cell proliferation in CNS-draining lymph nodes during chronic brain infection (*Figure 6a–d*). Interestingly, though, a large population of Ki67-expressing CD4+ and CD8+ T cells was also detected in the ILNs at this time point, albeit at a lower frequency (*Figure 6a–d*). These data suggest that T cell proliferation is enriched in lymph nodes that drain CNS tissue at 6 wpi, when parasite is most abundant in the brain, but also occurs in lymph nodes that do not drain CNS tissue.

To more closely examine the locations of antigen stimulation during chronic infection, we adoptively transferred in vitro-expanded OVA-specific OT-I cells (CD45.1 congenic) to C57BL/6 mice 5–6 weeks after infection with either Pru-OVA or parental strain Pru (*Figure 6e*). We then measured CellTrace Violet dilution in the DCLNs and ILNs 1 week after transfer. As expected, OT-I cells only accumulated in the brains of mice infected with Pru-OVA (*Figure 6f–g*), and all of the OT-I cells detected in the brains of these mice had fully diluted their CellTrace Violet dye (*Figure 6h*), consistent with the notion that these cells undergo several rounds of proliferation prior to recruitment to the brain.

To understand where peripheral T cells undergo activation and proliferation, we started by measuring CellTrace Violet dilution in the CNS-draining deep cervical lymph nodes. Notably, 46% of OT-I cells displayed complete dilution in the DCLNs when mice were infected with Pru-OVA, compared to only 11% of OT-I cells when mice were infected with parental Pru (*Figure 6i–j*). These data indicate that proliferation of SIINFEKL-specific CD8$^+$ T cells in CNS-draining lymph nodes is predominantly antigen-dependent, and only a limited degree of proliferation can be attributed to bystander activation. The OT-I cells used for adoptive transfer also expressed GFP under the control of the *Nr4a1* (Nur77) promoter, a reporter system that was originally developed as a tool to track antigen-dependent TCR stimulation (*Moran et al., 2011*; *Ashouri and Weiss, 2017*). However, we observed that GFP upregulation was equivalent in OT-I cells after infection with either Pru-OVA or parental Pru (*Figure 6—figure supplement 1a-d*), even though OT-I cells proliferated to a significantly higher degree in the lymph nodes of mice infected with Pru-OVA compared to mice infected with parental Pru (*Figure 6i–l*). To determine whether antigen-dependent T cell stimulation occurred in lymph nodes that drain non-CNS tissue during chronic infection, we measured CellTrace Violet dilution in the inguinal lymph nodes. Intriguingly, the degree of antigen-dependent T cell proliferation at this site was similar to that observed in the deep cervical lymph nodes, as 51% of OT-I cells fully diluted the tracer dye in mice infected with Pru-OVA compared to 8% of OT-I cells in mice infected with parental Pru (*Figure 6k–l*). These data lead us to conclude that antigenic stimulation of T cells occurs concurrently in CNS- and non-CNS-draining lymph nodes during chronic infection. What remains uncertain is the source of antigen eliciting T cell proliferation in the non-CNS-draining lymph nodes during chronic infection, though residual antigen persisting after acute infection seems a likely possibility due to the minimal presence of parasite in non-CNS tissues during the chronic stage of infection (*Figure 2f*).

In summary, these experiments reveal that both CNS-draining lymph nodes and non-CNS-draining lymph nodes are sources of newly activated parasite-specific T cells during the chronic stage of *T. gondii* infection. Ongoing proliferation of T cells in lymph nodes that do not drain the CNS, such as the inguinal lymph nodes, may explain the durability of T cell responses in the brain when T cell responses in the deep cervical lymph nodes become impaired.

## Discussion

Due to the unique anatomic organization of the CNS (*Plog and Nedergaard, 2018*; *Louveau et al., 2015a*), notably the absence of lymphatic vessels in the brain parenchyma, it has remained an open question how T cells outside the CNS detect and respond to microbial antigen expressed in the brain. The recent discovery of cerebrospinal fluid-draining lymphatic vessels in the dural meninges of mice and humans (*Louveau et al., 2015b*; *Absinta et al., 2017*) has challenged previous frameworks for how foreign antigen exits the CNS (*Walter et al., 2006*; *Goldmann et al., 2006*; *Weller et al., 2010*). Here, we provide evidence that meningeal lymphatic drainage supports dendritic cell activation and peripheral CD4$^+$ and CD8$^+$ T cell responses against the brain-tropic pathogen *T. gondii*. We further demonstrate that meningeal lymphatic drainage is dispensable for T cell responses against the parasite in the brain, despite reports showing drainage to be essential for the development of CNS-directed T cell responses in models of brain cancer (*Song et al., 2020*; *Hu et al., 2020*).

The deep cervical lymph nodes have been strongly implicated in mediating immune responses to CNS antigen (*Song et al., 2020*; *Louveau et al., 2018b*; *Harling-Berg et al., 1989*). Consistent with this notion, infection with the Pru-OVA strain of *T. gondii* led to robust expansion of SIINFEKL-specific T cells in the deep cervical lymph nodes, and the magnitude of this response tracked with parasite burden in the brain as infection progressed. In peripheral organs, foreign antigen is transported directly from the site of infection to lymph nodes for immune cell activation (*Liao and von der Weid, 2015*). However, our data support a model in which antigen expressed in infected brain tissue must first reach the cerebrospinal fluid before it can be transported to lymph nodes for T cell activation.

Indeed, parasite-specific T cell responses were significantly impaired in the DCLNs when lymphatic drainage of CSF components was restricted by surgical ligation. Future studies will be needed to more closely examine how CNS antigen is elaborated into the CSF.

In addition to the drainage of soluble antigen, lymphatic vessels promote the trafficking of dendritic cells from infected tissue after antigen uptake and phenotypic maturation (*Russo et al., 2013*). Even though the dural meninges did not harbor active infection at 6 wpi, we observed a significant accumulation of type 1 and type 2 conventional dendritic cells (cDC) at this site, with a large proportion of CD11c$^+$ antigen-presenting cells distributing specifically around the dural sinuses where they could sample CSF-derived protein. Interestingly, as infection progressed from the acute to chronic stage, there was a shift in the relative proportion of cDC1s and cDC2s, and cDC2s displayed a greater degree of activation. While it is well-established that cDC1s are critical for initiating Th1 immunity against *T. gondii* during acute infection (*Poncet et al., 2019*), less is known about the role of cDC1s and cDC2s during chronic infection. It also remains to be determined what factors released in the brain, CSF, or meninges signal on cDC2s to induce maturation during infection. Likely candidates include IFN-γ and TNF-α, which become highly expressed in the brain (*Matta et al., 2021*), as well as pathogen-associated molecular patterns, such as the TLR11 agonist *T. gondii* profilin (*Yarovinsky et al., 2005*), or damage-associated molecular patterns, which can accumulate in the CSF during *T. gondii* infection (*Still et al., 2020*). Even though CD11c$^{hi}$MHCII$^{hi}$ cells are observed to accumulate in the brain during *T. gondii* infection, it is unclear whether these cells are able to migrate to peripheral lymph nodes, a function that dural dendritic cells have been shown to exhibit (*Louveau et al., 2018b*). Some studies have suggested that dendritic cells in the brain travel along the rostral migratory stream and exit by crawling along olfactory nerves (*Mohammad et al., 2014*; *Kaminski et al., 2012*), but two-photon live imaging studies of *T. gondii*-infected mouse brains has revealed that CD11c$^+$ cells, including tissue-resident macrophages and dendritic cells, are largely immotile (*John et al., 2011*). EAE studies suggest that the function of dendritic cells in the CNS may be to support local reactivation of T cells or promote T cell entry into the CNS (*Goverman, 2009*; *D'Agostino et al., 2012*; *Mundt et al., 2019*). Thus, dendritic cells in the different anatomic compartments of the CNS likely play distinct roles in regulating immune responses to CNS antigen. Further studies are needed to clarify the function of dendritic cells at these sites, though challenges remain due to limitations in targeting cells within different compartments of the CNS.

Disrupting meningeal lymphatic drainage during chronic brain infection caused an impairment in parasite-specific T cell activation and proliferation in the DCLNs, but did not affect T cell responses in the brain. Because the T cell population in the brain requires continual replenishment by circulating T cells (*Harris et al., 2012*; *Wilson et al., 2009*), these results suggest that alternative sources of antigen likely exist outside the CNS-draining lymph nodes during chronic infection. In fact, we observed significant antigen-dependent proliferation of OT-I cells in both the deep cervical lymph nodes and the inguinal lymph nodes six weeks after infection. The source of antigen in the inguinal lymph nodes is still not completely clear. However, considering the robust T cell responses and high levels of parasite gDNA observed in the inguinal lymph nodes during acute infection, it is possible that residual antigen persisting after clearance of acute infection contributed to the proliferation of OT-I cells at this site during chronic infection. In models of influenza or vesicular stomatitis virus infection, antigen depots were found to persist for weeks to months after viral clearance (*Turner et al., 2007*; *Zammit et al., 2006*). It is also possible that low-grade infection of skeletal muscle (*Jin et al., 2017*; *Melchor et al., 2020*) contributes to peripheral T cell responses during chronic infection, although parasite gDNA was not detected in skeletal muscle by real-time PCR 6 weeks after infection with Pru-OVA. Studies examining T cell responses against bradyzoite-specific antigens could help clarify two important points: (1) the degree to which T cell activation in lymph nodes that drain non-CNS tissues is a result of chronic infection of these tissues and (2) the potential importance of CNS lymphatic drainage in generating immune responses to antigens specifically expressed by the latent form of the parasite.

The presence of peripheral antigen during *T. gondii* brain infection is not surprising, as most CNS pathogens display varying degrees of tropism for multiple tissues and originate at a peripheral site prior to hematogenous dissemination to the brain, CSF, or meninges (*Cain et al., 2019*). In fact, peripheral infection may serve to 'immunize' the host ahead of CNS infection, making the drainage of CNS antigen to the deep cervical lymph nodes more redundant. By contrast, when tumors originate within the CNS, meningeal lymphatic drainage was found to be indispensable for T cell-dependent

immunity in the brain (*Song et al., 2020*; *Hu et al., 2020*), likely because there were not alternative sources of antigen outside the CNS. Consistent with this idea, when tumor cells were injected into the flank of mice at the same time as intracranial engraftment, survival benefits of VEGF-C-enhanced meningeal lymphatic drainage were abrogated (*Song et al., 2020*). These results harken back to early studies performed by Medawar in the 1940s showing that skin grafts in the brain were rejected more quickly and at higher rates when also engrafted to the chest of rabbits (*Medawar, 1948*). For these reasons, we predict that meningeal lymphatic drainage may not be necessary to generate T cell responses against metastatic brain tumors, given the presence of early peripheral antigen prior to seeding of the brain.

To conclude, our study provides novel insight into how T cells respond to antigen expressed in the brain by a neurotropic pathogen. Our data shed new light on how the dural meninges promotes neuroimmune communication, while refining our understanding of how meningeal lymphatic drainage helps to preserve the health of the CNS.

## Methods

### Mice

All mice were housed at University of Virginia specific pathogen-free facilities with a 12 hr light/dark cycle. C57BL/6 J (#000664), CBA/J (#000656), B6.Cg-Tg(Itgax-Venus)1Mnz/J (*CD11c*$^{YFP}$ mice, #008829), C57BL/6-Tg(TcraTcrb)1100Mjb/J (OT-I mice, #003831), and C57BL/6-Tg(Nr4a1-EGFP/cre)820Khog/J (*Nr4a1*$^{GFP}$ mice, #016617) strains were originally purchased from the Jackson Laboratory and then maintained within our animal facility. B6.SJL-*Ptprc*$^a$*Pepc*$^b$/BoyCrCrl (CD45.1 congenic mice, #564) and Swiss Webster (#024) strains were purchased from Charles River Laboratories. To generate OT-I (CD45.1 congenic) mice, OT-I and CD45.1 congenic strains were cross-bred within our facility. OT-I (CD45.1 congenic) mice were then crossed to *Nr4a1*$^{GFP}$ mice to generate OT-I/*Nr4a1*$^{GFP}$ (CD45.1 congenic) mice for use in adoptive transfer studies. For experiments reported in this study, age-matched young adult female mice (7–10 weeks of age) were used. All experiments were approved by the Institutional Animal Care and Use Committee at the University of Virginia under protocol number 3968.

### Parasite strains and infection

Mice were infected with avirulent, type II strains of *T. gondii*. The ME49 strain was maintained in chronically infected (2–6 months) Swiss Webster mice and passaged through CBA/J mice. For experimental infections with the ME49 strain, tissue cysts were prepared from homogenized brains of chronically infected (4–8 weeks) CBA/J mice. Mice were then inoculated intraperitoneally (i.p.) with 10 tissue cysts of ME49 in 200 µl of 1 X PBS. The transgenic Prugniaud strain of *T. gondii* expressing ovalbumin (aa 140–386) and TdTomato (Pru-OVA) were generously provided by Anita Koshy (University of Arizona) and maintained by serial passage through human foreskin fibroblast (HFF) monolayers in parasite culture medium (DMEM [Gibco], 20% Medium 199 [Gibco], 10% FBS [Gibco], 1% penicillin/streptomycin [Gibco], and 10 µg/ml gentamicin [Gibco]). The parental Prugniaud strain (Pru$^{ΔHPT}$, parental Pru) was similarly maintained. For experimental infections with the Pru-OVA or parental Pru strains, tachyzoites were purified from HFF cultures by needle-passaging scraped cells and filtering the parasites through a 5.0 µm filter (EMD Millipore). Mice were then inoculated i.p. with 1000 tachyzoites of Pru-OVA or parental Pru in 200 µl of 1 X PBS.

### Assessment of parasite burden

For assessment of parasite burden by real-time PCR, genomic DNA (gDNA) was isolated from mouse brains, dural meninges, quadriceps femoris skeletal muscle, lymph nodes, or subcutaneous adipose tissue using the Isolate II Genomic DNA Kit (Bioline, BIO-52067) following the manufacturer's instructions. Prior to tissue lysis, whole brains and skeletal muscle required mechanical homogenization using an Omni TH tissue homogenizer (Omni International). Amplification of the 529 bp repeat element in the *T. gondii* genome was performed as described previously (*Homan et al., 2000*), using the *Taq* polymerase-based SensiFAST Probe No-ROX Kit (Bioline, BIO-86005) and CFX384 Real-Time System (Bio-Rad) to assay 500 ng of DNA per sample. A standard curve generated from 10-fold serial dilutions of *T. gondii* gDNA, isolated from cultured HFFs and ranging from 3 to 300,000 genome copies, was

used to determine the total number of *T. gondii* genome copies per µg gDNA per tissue. For assessment of parasite burden by cyst counts, whole brains were minced with a razor blade and passed through an 18-gauge and 22-gauge needle to mechanically homogenize the tissue. A total of 30 µl of tissue homogenate was then mounted on a microscope slide and *T. gondii* cysts were enumerated using a DM 2000 LED brightfield microscope (Leica).

## Intra-cisterna magna injections

Mice were anesthetized by i.p. injection of a solution containing ketamine (100 mg/kg) and xylazine (10 mg/kg) diluted in saline. Mice were secured in a stereotaxic frame with the head angled slightly downward. An incision in the skin was made at the base of the skull and the muscle layers overlying the atlanto-occipital membrane were retracted. A 33-gauge Hamilton syringe (#80308) was inserted at a steep angle through the membrane to inject the desired solution into the CSF-filled cisterna magna. After injection, the skin was closed using 5–0 nylon sutures and mice received a subcutaneous (s.c.) injection of ketoprofen (2 mg/kg). Mice were then allowed to recover on a heating pad until awake.

## Lymphatic vessel ligation

Collecting lymphatic vessels afferent to the deep cervical lymph nodes were surgically ligated as previously described (*Louveau et al., 2015b*). Mice were anesthetized by i.p. injection of a solution containing ketamine (100 mg/kg) and xylazine (10 mg/kg) diluted in saline. A midline incision was made into the skin overlying the anterior neck after being shaved and cleaned with povidone-iodine and 70% ethanol. The sternocleidomastoid (SCM) muscles were retracted and collecting lymphatic vessels afferent to the DCLNs were ligated using 9–0 nylon suture (Living Systems Instrumentation, THR-G). After bilateral ligation of the lymphatic vessels, the skin was closed using 5–0 nylon sutures and mice received s.c. injection of ketoprofen (2 mg/kg). Mice were then allowed to recover on a heating pad until awake. Sham operations were performed in a similar manner, except lymphatic vessels were not ligated after retraction of the SCM muscles and exposure of the DCLNs. In one experiment (*Figure 5—figure supplement 4*), ligation of lymphatic vessels draining cerebrospinal fluid (CSF) to the superficial cervical lymph nodes (SCLN) was also performed. Because not all lymphatic vessels afferent to the SCLNs transport CSF, Evans blue tracer studies were completed to identify the six afferent lymphatic vessels (three on each side) that carry CSF to the SCLNs. These lymphatic vessels were bilaterally ligated using 9–0 nylon suture immediately after ligation of the lymphatic vessels afferent to the DCLNs and just prior to wound closure.

## Cerebrospinal fluid tracer analysis

To assess the function of meningeal lymphatic drainage in mice that had received ligation surgery, tracers were injected into the cerebrospinal fluid (CSF) and outflow to the deep cervical lymph nodes or superficial cervical lymph nodes was measured. For qualitative assessment, 5 µl of 10% Evans blue (Sigma-Aldrich) diluted in artificial CSF (aCSF, Harvard Apparatus) was introduced into the subarachnoid space of ligated or sham-operated mice by intra-cisterna magna (i.c.m.) injection. One hour after injection, mice were sacrificed and drainage of the dye to the DCLNs was visualized under a S6D stereomicroscope (Leica). For quantitative assessment, 3 µl of Alexa Fluor 594-conjugated ovalbumin (OVA-AF594, Thermo Fisher) or Alexa Fluor 488-conjugated ovalbumin (OVA-AF488, Thermo Fisher) diluted in aCSF (2 mg/ml) was introduced into the subarachnoid space of ligated or sham-operated mice by i.c.m. injection. Two hours after injection, mice were sacrificed and the amount of OVA-AF594 or OVA-AF488 present in the DCLNs or SCLNs was determined by fluorescence spectroscopy. Briefly, DCLNs and SCLNs were harvested and enzymatically digested with Collagenase D (1 mg/ml, Sigma-Aldrich) at 37 °C for 30 min, followed by a 2 hr treatment in T-PER Tissue Protein Extraction Reagent (Thermo Scientific) at 4 °C. Samples were centrifuged at >10,000 rpm to remove cellular debris and the supernatant containing extracted protein was transferred to a black flat-bottom microwell plate (Millipore Sigma, M9685). Relative fluorescence intensity of each sample was measured on a SpectraMax iD3 microplate reader (Molecular Devices) using an excitation wavelength of 590 nm and emission wavelength of 630 nm for OVA-AF594 and an excitation wavelength of 490 nm and emission wavelength of 530 nm for OVA-AF488 before comparison to a standard curve to determine the total amount of tracer that drained.

## DQ-OVA analysis

The uptake and processing of CSF-derived protein by antigen-presenting cells outside the CNS was evaluated by injecting 3 µl of DQ-OVA (2 mg/ml, Invitrogen) into the CSF of chronically infected C57BL/6 mice by i.c.m. injection. The fluorescent signal of proteolytically cleaved DQ-OVA, with a peak excitation wavelength of 505 nm and peak emission wavelength of 515 nm, was then detected by flow cytometry or confocal microscopy. For flow cytometry studies, the dural meninges, deep cervical lymph nodes, and inguinal lymph nodes were harvested and prepared as single-cell suspensions for cell surface staining 2 hr, 5 hr, or 24 hr after i.c.m. injection of DQ-OVA. For flow acquisition, fluorescence emitted by processed DQ-OVA was detected using the FL1 sensor (525/40 nm) of a Gallios flow cytometer (Beckman Coulter) or using a Cytek Aurora Flow Cytometry System. For imaging studies, mice were sacrificed 12 h after injection of DQ-OVA, and whole-mount dural meninges were fixed, then immunostained using directly-conjugated antibodies targeting MHC class II (Super Bright 436, eBioscience) and LYVE1 (eFluor 660, eBioscience). Fluorescence emitted by processed DQ-OVA was detected using 488 nm laser excitation on a TCS SP8 confocal microscope (Leica).

## Immunohistochemistry

Image analysis was performed on dural meninges whole mounts or on sectioned brain tissue. After sacrifice, transcardiac perfusion of mice was performed using 20 ml of cold 1 X PBS. The dorsal aspect of the skull was removed and the dural meninges were fixed, while still attached to the skull bone, in 4% PFA for 6–8 hr at 4 °C. Then, as previously described (*Louveau et al., 2018a*), whole-mount dural meninges were carefully dissected and stored in 1 X PBS. To prepare brains for image analysis, mice were perfused with 20 ml of cold 4% PFA immediately after perfusion with 1 X PBS. Brains were bisected along the sagittal midline and post-fixed in cold 4% PFA for 24 hr. Brains were then cryoprotected in 30% sucrose for 24 hr at 4 °C, embedded in OCT (Tissue Tek), and frozen on dry ice. 50-µm sections were then prepared using a CM 1950 cryostat (Leica) and stored in 1 X PBS as free-floating sections. To immunostain whole mount meninges or free-floating brain sections, the fixed tissues were first incubated in a blocking solution (2% normal donkey serum, 1% BSA, 0.05% Tween 20, and 0.5% Triton X-100 in 1 X PBS) at room temperature for 45 min. Then, tissue was stained for 2 hr at room temperature or overnight at 4 °C with directly-conjugated primary antibodies. If a secondary antibody was used, then samples were washed three times in a solution of 0.05% Tween 20 (in 1 X PBS) and stained for 1 hr at room temperature. Finally, tissues were washed three times in a solution of 0.05% Tween 20 (in 1 X PBS), mounted onto glass slides using AquaMount (Fisher Scientific), and coverslipped. In some experiments, the tissue was counter-stained with DAPI (Thermo Scientific) and washed just before mounting onto slides. Primary antibodies (from eBioscience) included: MHC class II (I-A/I-E)-Super Bright 436 (62-5321-80), LYVE1-EF570 (41-0443-82), LYVE1-EF660 (50-0443-82), and anti-ME49 (gift from Fausto Araujo). Secondary antibody (used to stain ME49): donkey anti-rabbit-AF488 (Thermo Fisher). Images were acquired using a Leica TCS SP8 confocal microscope and analyzed using Fiji software (*Schindelin et al., 2012*). Experimenter was blinded to the identity of experimental group during image analysis.

## Tissue processing for flow cytometry

After sacrifice, transcardiac perfusion of mice was performed using 20 ml of cold 1 X PBS. For analysis of the dural meninges, the dorsal aspect of the skull was removed and the dural meninges were carefully dissected from the skull bone under a S6D stereomicroscope (Leica). The tissue was then digested in 1 X HBSS (without $Ca^{2+}$ or $Mg^{2+}$, Gibco) with Collagenase D (1 mg/ml, Sigma-Aldrich) and Collagenase VIII (1 mg/ml, Sigma-Aldrich) at 37 °C for 30 min, before being passed through a 70 µm strainer (Corning). Cells were then pelleted, resuspended, and kept on ice. Deep cervical lymph nodes (DCLNs), which are positioned laterally to the trachea and underneath the sternocleidomastoid muscles, and inguinal lymph nodes (ILNs) were harvested bilaterally into cold complete RPMI media (cRPMI; 10% FBS [Gibco], 1% penicillin/streptomycin [Gibco], 1% sodium pyruvate [Gibco], 1% non-essential amino acids [Gibco], and 0.1% 2-Mercaptoethanol [Life Technologies]). Lymph nodes were mechanically homogenized and gently pressed through a 70 µm strainer (Corning). Cells were then pelleted, resuspended, and kept on ice. Brains were harvested into cold cRPMI, minced with a razor blade, and passed through an 18-gauge and 22-gauge needle for mechanical homogenization. Tissue was then digested in a solution containing collagenase/dispase (0.227 mg/ml, Sigma-Aldrich)

and DNase (50 U/ml, Roche) at 37 °C for 45–60 min, before being passed through a 70 µm strainer (Corning) and washed with cRPMI. Myelin was separated from mononuclear cells by resuspending samples in 20 ml of 40% Percoll (Cytiva) and centrifuging at 650 g for 25 min. Myelin was aspirated and cell pellets were washed, resuspended, and kept on ice. Spleens were harvested into cold cRPMI, then mechanically homogenized and washed through a 40 µm strainer (Corning). Cells were resuspended in RBC lysis solution (0.16 M NH$_4$Cl) for 2 min. Cells were then washed, resuspended, and kept on ice.

## Cerebrospinal fluid collection for flow cytometry

Mice were anesthetized by i.p. injection of a solution containing ketamine (100 mg/kg) and xylazine (10 mg/kg) diluted in saline. An incision in the skin was made at the base of the skull and the muscle layers overlying the atlanto-occipital membrane were retracted. A pulled glass capillary (Sutter Instrument, BF100-50-10) was inserted through the membrane into the cisterna magna and 5–10 µl of CSF was collected per mouse. CSF from four to five mice was pooled, and a total of 25 µl of CSF per pooled sample was analyzed by flow cytometry.

## Flow cytometry

Single-cell suspensions prepared from tissues or cerebrospinal fluid were pipetted into a 96-well plate and pelleted. Eight-color experiments were performed on a Gallios flow cytometer (Beckman Coulter) and prepared as follows: Cells were treated with 50 µl of Fc block (0.1% rat gamma globulin [Jackson ImmunoResearch], 1 µg/ml of 2.4G2 [BioXCell]) for 10 min at room temperature. Cells were then stained for surface markers and incubated with a fixable live/dead viability dye for 30 min at 4 °C. In experiments where SIINFEKL-specific CD8$^+$ T cells were analyzed, cells were pre-incubated with PE-conjugated H-2K$^b$/OVA (SIINFEKL) tetramer (NIH Tetramer Core Facility) for 15 min at room temperature prior to cell surface staining. After surface staining, cells were washed with FACS buffer (0.2% BSA and 2 mM EDTA in 1 X PBS) and, if intracellular staining was performed, cells were treated with a fixation/permeabilization solution (eBioscience, 00-5123-43 and 00-5223-56) overnight then stained for intracellular markers in permeabilization buffer (eBioscience, 00-8333-56) for 30 min at 4 °C. Finally, samples were resuspended in FACS buffer and acquired. Fourteen-color experiments were performed using a Cytek Aurora Flow Cytometry System (spectral flow cytometry) and prepared as follows: Cells were stained with a Live/Dead Fixable Red Dead Cell Stain (Thermo Fisher, L34971) according to the manufacturer's instructions. Cells were washed and then treated with 50 µl of Fc block (as above) for 10 min at room temperature. Cells were then stained for surface markers for 30 min at 4 °C. In experiments where SIINFEKL-specific CD8$^+$ T cells were analyzed, cells were pre-incubated with PE-conjugated H-2K$^b$/OVA (SIINFEKL) tetramer (NIH Tetramer Core Facility) for 15 min at room temperature prior to cell surface staining. Finally, samples were resuspended in FACS buffer and acquired after spectral unmixing using single-color controls. Samples acquired on the Gallios flow cytometer and Cytek Aurora Flow Cytometry System were analyzed using FlowJo software v.10. For cerebrospinal fluid, cell counts were determined using absolute counting beads (Life Technologies, C36950) pipetted into samples just prior to acquisition. For lymph nodes, brain, and dural meninges, cell counts were measured using a Hausser Scientific hemacytometer (Fisher Scientific). For samples analyzed using the Gallios flow cytometer, cells were stained for surface markers using the following eBioscience antibodies at 1:200 dilution: CD45-FITC (11-0451-82), CD62L-FITC (11-0621-85), CD80-FITC (11-0801-82), MHC class II (I-A/I-E)-PE (12-5321-82), CD69-PE (12-0691-82), CD11c-PerCP-Cy5.5 (45-0114-82), CD4-PerCP-Cy5.5 (45-0042-82), CD45-PerCP-Cy5.5 (45-0451-82), CD11b-PerCP-Cy5.5 (45-0112-82), CD8α-PerCP-Cy5.5 (45-0081-82), CD45.1-PerCP-Cy5.5 (45-0453-82), CD80-PE-Cy7 (25-0801-82), CD11c-PE-Cy7 (25-0114-82), CD4-PE-Cy7 (25-0041-82), CD8α-PE-Cy7 (25-0081-82), TCR-β-APC (17-5961-82), NK1.1-APC (17-5941-82), CD19-APC (17-0193-82), CD44-AF780 (47-0441-82), CD11b-AF780 (47-0112-82), CD86-EF450 (48-0862-82), MHC class II (I-A/I-E)-EF450 (48-5321-82), CD69-EF450 (48-0691-82), and CD4-EF450 (48-0042-82). Cells were stained for intracellular markers using the following eBioscience antibodies at 1:200 dilution: TNF-α-AF488 (53-7321-82), IFN-γ-PerCP-Cy5.5 (45-7311-82), and Ki67-PE-Cy7 (25-5698-82). The following eBioscience live/dead dyes were used at a 1:800 dilution: Fixable Viability Dye eFluor 780 (65-0865-14) and Fixable Viability Dye eFluor 506 (65-0866-18). For samples analyzed using the Cytek Aurora Flow Cytometry System, cells were stained for surface markers using the following antibodies at 1:200 dilution (except where indicated):

CD45-AF700 (BioLegend, 103127), CD11b-BB700 (BD Horizon, 566417), CD64-PE-Cy7 (BioLegend, 139314), CD19-BV711 (BD Horizon, 563157), CD3e-BV711 (BD Horizon, 563123), NK1.1-BV711 (BioLegend, 108745), Ly6G-BV711 (BioLegend, 127643), TER119-BV711 (BD OptiBuild, 740686), CD11c-EF506 (Invitrogen, 69-0114-80), MHC II (I-A/I-E)-SB780 (eBioscience, 78-5321-82), CD26-PE (BioLegend, 137803), XCR1-APC (BioLegend, 148205), SIRPα-SB436 (Invitrogen, 62-1721-82) (used at 1:100 dilution), CD103-BV605 (BioLegend, 121433), CD80-BV650 (BD Horizon, 563687), CD86-APC-EF780 (Invitrogen, 47-0862-80), TCR-β-APC (Invitrogen, 17-5961-82), CD4-BV650 (BD Horizon, 563232), CD8α-BV421 (BD Horizon, 563898), CD45.1-BV711 (BioLegend, 110739), PD1-BV786 (BD OptiBuild, 744548), TIM3-APC-H7 (BD Pharmingen, 567165), LAG3-SB600 (Invitrogen, 63-2231-82), CXCR3-PE-Cy7 (BioLegend, 126515), KLRG1-EF506 (Invitrogen, 69-5893-80), CD69-AF700 (BD Pharmingen, 561238), and CD103-PerCP-EF710 (Invitrogen, 46-1031-80).

### Ex vivo T cell restimulation

Cells isolated from the deep cervical lymph nodes or brain were seeded at $2.5 \times 10^5$ cells per well and restimulated for 24 hr with soluble tachyzoite antigen (STAg) at 25 µg/ml. Cells were then incubated with brefeldin A (10 µg/ml, Selleck Chemicals) at 37 °C for 6 hr. Cells were washed and stained for surface markers, and then intracellular cytokine staining was performed. To prepare STAg, tachyzoites of the RH strain of *T. gondii* were purified from HFF cultures, filtered through a 5.0 µm filter (EMD Millipore), resuspended in 1 X PBS, and lysed by five freeze-thaw cycles. Protein concentration was determined by BCA assay (Pierce) and stock solutions were stored at –80 °C.

### OT-I cell labeling and transfer

Spleens and lymph nodes from OT-I/*Nr4a1*$^{GFP}$ (CD45.1 congenic) mice or OT-I (CD45.1 congenic) were pooled, mechanically dissociated, treated with RBC lysis solution (0.16 M NH$_4$Cl), and passed through a 70 µm strainer (Corning). Then, OT-I cells were expanded in vitro as previously described (*Harris et al., 2012*). Briefly, isolated cells were cultured for 24 hr with ovalbumin protein (Worthington, LS003048) at 500 µg/ml. After 24 hr, cells were washed and rested. On days 4 and 6, cells were treated with recombinant IL-2 (Proleukin, Prometheus Laboratories) at 200 U/ml. On day 7, cells were resuspended in 1 X PBS and then $4 \times 10^6$ to $5 \times 10^6$ labeled OT-I cells were transferred to mice anesthetized with ketamine (100 mg/kg) and xylazine (10 mg/kg) by retro-orbital intravenous injection. For dye dilution experiments, cells were labeled with 10 µM CellTrace Violet (Invitrogen, C34557) for 20 min at 37 °C prior to adoptive transfer.

### Statistical analysis

Statistical analyses were performed using Prism software (v8.4) or RStudio (v1.1) statistical packages. Power analysis was performed in R using the pwr software package to calculate group sizes needed to achieve a power of 0.8, with effect size estimated from preliminary experiments. Randomization of samples was ensured by including mice from the same experimental group in different cages. A two-tailed student's *t*-test was used to compare two experimental groups. A one-way ANOVA was performed to compare three or more experimental groups with a single independent variable. A two-way ANOVA was performed to compare three or more experimental groups with two independent variables. When data were pooled from multiple experiments, a randomized block ANOVA was performed using the lme4 software package in R (*Bates, 2015*). This test models experimental groups as a fixed effect and experimental day as a random effect. The test used for each experiment is denoted in the figure legend, and p values are similarly indicated, with ns = not significant, p<0.05 (*), p<0.01 (**), and p<0.001 (***). Graphs were generated using Prism software and show mean values ± s.e.m. along with individual data points representative of individual mice (biological replicates).

## Acknowledgements

We acknowledge members of the Center for Brain Immunology and Glia (BIG) at the University of Virginia for their scientific input, training in surgical techniques, and access to instrumentation as this project was being developed. Schematic diagrams were generated using BioRender (https://biorender.com/). We thank Marieke K Jones for her guidance with statistical analyses and R programming. We thank Anita Koshy at the University of Arizona for providing transgenic parasite strains used in this study. We thank Fausto Araujo at Palo Alto Medical Foundation for gifting us the rabbit

anti-ME49 antibody used in this study. We also acknowledge the support we received from the Biomolecular Analysis Facility at the University of Virginia. This work was funded by National Institute of Health grants R01NS112516, R01NS091067, R21NS128551, and R56NS106028 to THH; F30AI154740, 5T32AI007496 and 5T32GM007267 to MAK; T32AI007496 to MNC and IWB; T32GM008328 to KS; and T32AI007046 to SJB. This work was also funded by the University of Virginia Pinn Scholars Award.

## Additional information

### Funding

| Funder | Grant reference number | Author |
|---|---|---|
| National Institutes of Health | R01NS112516 | Tajie H Harris |
| National Institutes of Health | R21NS128551 | Tajie H Harris |
| National Institutes of Health | F30AI154740 | Michael A Kovacs |
| National Institutes of Health | T32AI007496 | Michael A Kovacs Maureen N Cowan Isaac W Babcock |
| National Institutes of Health | T32GM007267 | Michael A Kovacs |
| National Institutes of Health | T32AI007046 | Samantha J Batista |
| National Institutes of Health | T32GM008328 | Katherine Still |
| National Institutes of Health | R01NS091067 | Tajie H Harris |
| National Institutes of Health | R56NS106028 | Tajie H Harris |
| University of Virginia | Pinn Scholars Award | Tajie H Harris |

The funders had no role in study design, data collection and interpretation, or the decision to submit the work for publication.

### Author contributions

Michael A Kovacs, Conceptualization, Data curation, Formal analysis, Funding acquisition, Validation, Investigation, Visualization, Methodology, Writing - original draft, Writing – review and editing; Maureen N Cowan, Isaac W Babcock, Lydia A Sibley, Conceptualization, Investigation, Writing – review and editing; Katherine Still, Samantha J Batista, Conceptualization, Investigation; Sydney A Labuzan, Investigation, Writing – review and editing; Ish Sethi, Investigation; Tajie H Harris, Supervision, Funding acquisition, Project administration, Writing – review and editing

### Author ORCIDs

Michael A Kovacs http://orcid.org/0000-0002-4298-9609
Tajie H Harris http://orcid.org/0000-0002-1355-2109

### Ethics

All experiments were approved by the Institutional Animal Care and Use Committee at the University of Virginia under protocol number 3968. When surgeries were performed on mice, mice were anesthetized using a solution containing ketamine (100 mg/kg) and xylazine (10 mg/kg) diluted in saline, and to minimize pain post-surgery mice were treated with ketoprofen (2 mg/kg).

### Decision letter and Author response

Decision letter https://doi.org/10.7554/eLife.80775.sa1
Author response https://doi.org/10.7554/eLife.80775.sa2

## Additional files

### Supplementary files
• MDAR checklist

### Data availability
All data generated and analyzed during this study are included in the manuscript and supporting figures. Source data has been provided for Figures 2c-f.

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

# Appendix 1

## Appendix 1—key resources table

| Reagent type (species) or resource | Designation | Source or reference | Identifiers | Additional information |
|---|---|---|---|---|
| Strain, strain background (*Mus. musculus*) | C57BL/6 J | Jackson Laboratory | Stock #:000664 | |
| Strain, strain background (*Mus. musculus*) | CBA/J | Jackson Laboratory | Stock #:000656 | |
| Strain, strain background (*Mus. musculus*) | CD45.1 congenic; B6.SJL-*Ptprc*ᵃ*Pepc*ᵇ/BoyCrCrl | Charles River Laboratory | Stock #:564 | |
| Strain, strain background (*Mus. musculus*) | Swiss Webster | Charles River Laboratory | Stock #:024 | |
| Genetic reagent (*Mus. musculus*) | CD11c-YFP; B6.Cg-Tg(Itgax-Venus)1Mnz/J | Jackson Laboratory | Stock #:008829 | |
| Genetic reagent (*Mus. musculus*) | OT-I; C57BL/6-Tg(TcraTcrb)1100Mjb/J | Jackson Laboratory | Stock #:003831 | |
| Genetic reagent (*Mus. musculus*) | Nur77-GFP; C57BL/6-Tg(Nr4a1-EGFP/cre)820Khog/J | Jackson Laboratory | Stock #:016617 | |
| Genetic reagent (*Mus. musculus*) | OT-I (CD45.1 congenic) mice | This paper | | OT-I mice were crossed with CD45.1 mice |
| Genetic reagent (*Mus. musculus*) | OT-I/Nur77GFP (CD45.1 congenic) | This paper | | OT-I/CD45.1 mice were crossed with Nur77-GFP mice |
| Genetic reagent (*background* (*Toxoplasma gondii*)) | PruΔHPT | DOI: 10.1371/journal.ppat.1010296 | | Generously provided by Anita Koshy at the University of Arizona |
| Genetic reagent (*background* (*Toxoplasma gondii*)) | Pru-OVA-TdTomato | DOI: 10.1371/journal.ppat.1010296 | | Generously provided by Anita Koshy at the University of Arizona |
| Antibody | anti-MHC II-Super Bright 436 (rat monoclonal) | eBioscience | Cat. #:62-5321-82 | IHC (1:100) |
| Antibody | anti-LYVE1-EF570 (rat monoclonal) | eBioscience | Cat. #:41-0443-82 | IHC (1:100) |
| Antibody | anti-LYVE1-EF660 (rat monoclonal) | eBioscience | Cat. #:50-0443-82 | IHC (1:100) |
| Antibody | anti-ME49 (rabbit polyclonal) | Other | | IHC (1:10,000); generously provided by Fausto Araujo at Palo Alto Medical Foundation |
| Antibody | Anti-rabbit-AF488 (donkey polyclonal) | Thermo Fisher | Cat. #:A-21206 | IHC (1:500) |
| Antibody | anti-CD45-FITC (rat monoclonal) | eBioscience | Cat. #:11-0451-82 | FC (1:200) |
| Antibody | anti-CD62L-FITC (rat monoclonal) | eBioscience | Cat. #:11-0621-85 | FC (1:200) |
| Antibody | anti-CD80-FITC (hamster monoclonal) | eBioscience | Cat. #:11-0801-82 | FC (1:200) |
| Antibody | anti-MHC class II (I-A/I-E)-PE (rat monoclonal) | eBioscience | Cat. #:12-5321-82 | FC (1:200) |
| Antibody | anti-CD69-PE (hamster monoclonal) | eBioscience | Cat. #:12-0691-82 | FC (1:200) |
| Antibody | anti-CD11c-PerCP-Cy5.5 (hamster monoclonal) | eBioscience | Cat. #:45-0114-82 | FC (1:200) |
| Antibody | anti-CD4-PerCP-Cy5.5 (rat monoclonal) | eBioscience | Cat. #:45-0042-82 | FC (1:200) |
| Antibody | anti-CD45-PerCP-Cy5.5 (rat monoclonal) | eBioscience | Cat. #:45-0451-82 | FC (1:200) |

*Appendix 1 Continued on next page*

*Appendix 1 Continued*

| Reagent type (species) or resource | Designation | Source or reference | Identifiers | Additional information |
|---|---|---|---|---|
| Antibody | anti-CD11b-PerCP-Cy5.5 (rat monoclonal) | eBioscience | Cat. #:45-0112-82 | FC (1:200) |
| Antibody | anti-CD8α-PerCP-Cy5.5 (rat monoclonal) | eBioscience | Cat. #:45-0081-82 | FC (1:200) |
| Antibody | anti-CD45.1-PerCP-Cy5.5 (mouse monoclonal) | eBioscience | Cat. #:45-0453-82 | FC (1:200) |
| Antibody | anti-CD80-PE-Cy7 (hamster monoclonal) | eBioscience | Cat. #:25-0801-82 | FC (1:200) |
| Antibody | anti-CD11c-PE-Cy7 (hamster monoclonal) | eBioscience | Cat. #:25-0114-82 | FC (1:200) |
| Antibody | anti-CD4-PE-Cy7 (rat monoclonal) | eBioscience | Cat. #:25-0041-82 | FC (1:200) |
| Antibody | anti-CD8α-PE-Cy7 (rat monoclonal) | eBioscience | Cat. #:25-0081-82 | FC (1:200) |
| Antibody | anti-TCR-β-APC (hamster monoclonal) | eBioscience | Cat. #:17-5961-82 | FC (1:200) |
| Antibody | anti-NK1.1-APC (mouse monoclonal) | eBioscience | Cat. #:17-5941-82 | FC (1:200) |
| Antibody | anti-CD19-APC (rat monoclonal) | eBioscience | Cat. #:17-0193-82 | FC (1:200) |
| Antibody | anti-CD44-AF780 (rat monoclonal) | eBioscience | Cat. #:47-0441-82 | FC (1:200) |
| Antibody | anti-CD11b-AF780 (rat monoclonal) | eBioscience | Cat. #:47-0112-82 | FC (1:200) |
| Antibody | anti-CD86-EF450 (rat monoclonal) | eBioscience | Cat. #:48-0862-82 | FC (1:200) |
| Antibody | anti-MHC class II (I-A/I-E)-EF450 (rat monoclonal) | eBioscience | Cat. #:48-5321-82 | FC (1:200) |
| Antibody | anti-CD69-EF450 (hamster monoclonal) | eBioscience | Cat. #:48-0691-82 | FC (1:200) |
| Antibody | anti-CD4-EF450 (rat monoclonal) | eBioscience | Cat. #:48-0042-82 | FC (1:200) |
| Antibody | anti-TNF-α-AF488 (rat monoclonal) | eBioscience | Cat. #:53-7321-82 | FC (1:200) |
| Antibody | anti-IFN-γ-PerCP-Cy5.5 (rat monoclonal) | eBioscience | Cat. #:45-7311-82 | FC (1:200) |
| Antibody | anti-Ki67-PE-Cy7 (rat monoclonal) | eBioscience | Cat. #:25-5698-82 | FC (1:200) |
| Antibody | anti-CD45-AF700 (rat monoclonal) | BioLegend | Cat. #:103127 | FC (1:200) |
| Antibody | anti-CD11b-BB700 (rat monoclonal) | BD Biosciences | Cat. #:566417 | FC (1:200) |
| Antibody | anti-CD64-PE-Cy7 (mouse monoclonal) | BioLegend | Cat. #:139314 | FC (1:200) |
| Antibody | anti-CD19-BV711 (rat monoclonal) | BD Biosciences | Cat. #:563157 | FC (1:200) |
| Antibody | anti-CD3e-BV711 (hamster monoclonal) | BD Biosciences | Cat. #:563123 | FC (1:200) |
| Antibody | anti-NK1.1-BV711 (mouse monoclonal) | BioLegend | Cat. #:108745 | FC (1:200) |
| Antibody | anti-Ly6G-BV711 (rat monoclonal) | BioLegend | Cat. #:127643 | FC (1:200) |
| Antibody | anti-TER119-BV711 (rat monoclonal) | BD Biosciences | Cat. #:740686 | FC (1:200) |

*Appendix 1 Continued on next page*

*Appendix 1 Continued*

| Reagent type (species) or resource | Designation | Source or reference | Identifiers | Additional information |
|---|---|---|---|---|
| Antibody | anti-CD11c-EF506 (hamster monoclonal) | eBioscience | Cat. #:69-0114-80 | FC (1:200) |
| Antibody | anti-MHC II (I-A/I-E)-SB780 (rat monoclonal) | eBioscience | Cat. #:78-5321-82 | FC (1:200) |
| Antibody | anti-CD26-PE (rat monoclonal) | BioLegend | Cat. #:137803 | FC (1:200) |
| Antibody | anti-XCR1-APC (mouse monoclonal) | BioLegend | Cat. #:148205 | FC (1:200) |
| Antibody | anti-SIRPα-SB436 (rat monoclonal) | eBioscience | Cat. #:62-1721-82 | FC (1:100) |
| Antibody | anti-CD103-BV605 (hamster monoclonal) | BioLegend | Cat. #:121433 | FC (1:200) |
| Antibody | anti-CD80-BV650 (hamster monoclonal) | BD Biosciences | Cat. #:563687 | FC (1:200) |
| Antibody | anti-CD86-APC-EF780 (rat monoclonal) | eBioscience | Cat. #:47-0862-80 | FC (1:200) |
| Antibody | anti-CD4-BV650 (rat monoclonal) | BD Biosciences | Cat. #:563232 | FC (1:200) |
| Antibody | anti-CD8α-BV421 (rat monoclonal) | BD Biosciences | Cat. #:563898 | FC (1:200) |
| Antibody | anti-CD45.1-BV711 (mouse monoclonal) | BioLegend | Cat. #:110739 | FC (1:200) |
| Antibody | anti-PD1-BV786 (hamster monoclonal) | BD Biosciences | Cat. #:744548 | FC (1:200) |
| Antibody | anti-TIM3-APC-H7 (mouse monoclonal) | BD Biosciences | Cat. #:567165 | FC (1:200) |
| Antibody | anti-LAG3-SB600 (rat monoclonal) | eBioscience | Cat. #:63-2231-82 | FC (1:200) |
| Antibody | anti-CXCR3-PE-Cy7 (hamster monoclonal) | BioLegend | Cat. #:126515 | FC (1:200) |
| Antibody | anti-KLRG1-EF506 (hamster monoclonal) | eBioscience | Cat. #:69-5893-80 | FC (1:200) |
| Antibody | anti-CD69-AF700 (hamster monoclonal) | BD Biosciences | Cat. #:561238 | FC (1:200) |
| Antibody | anti-CD103-PerCP-EF710 (hamster monoclonal) | eBioscience | Cat. #:46-1031-80 | FC (1:200) |

