## [Editor Report]

Kovacs et al. provide important and valuable data on the role of meningeal lymphatic drainage in T cell responses during chronic *Toxoplasma gondii* (T.g.) infection in mice. They present compelling data showing dendritic cell (DC) accumulation in the dura and CSF at 6 weeks post-infection, which matches with the replication peak of T.g. in the brain, and with T cell expansion/activation in the draining lymph node (dCLN). They also convincingly show that during chronic infection, antigen-specific T cells are generated not only in the dCLN but also in the periphery (ILN), which could account for the presence of T cells in the brain after surgical blockade of the lymphatics. This study highlights also how the CNS is protected against environmental challenges.

---

## [Decision Letter]

**Decision letter after peer review:**

Thank you for submitting your article "Meningeal lymphatic drainage promotes T cell responses against *Toxoplasma gondii* but is dispensable for parasite control in the brain" for consideration by *eLife*. Your article has been reviewed by 2 peer reviewers, and the evaluation has been overseen by a Reviewing Editor and Betty Diamond as the Senior Editor. The following individual involved in the review of your submission has agreed to reveal their identity: Sarah Mundt (Reviewer #2).

Essential revision:

We all agree that the study warrants publication pending adequate revision.

Please find below the key revision points that need to be addressed.

1) A better gating strategy to quantify DC subsets with kinetics.

2) A better definition of T cell responses with kinetics and fate of OT1/ dCLN generated T cell generated in the chronic versus acute phase.

3) Kinetics of parasite replication in the periphery (e.g. in tissue draining to ILN) and in the dural meninges, for WT and/or Pru-OVA strains to provide a better context to understand the localization of T cell priming in chronic and acute phases.

*Reviewer #1 (Recommendations for the authors):*

1. In Figure 1, DC are quantified using CD11cYFP reporter, or CD11c gating strategy (which by the way is not very clean in several figures), which is not sufficient to prove that those are DCs (for instance, activated macrophages can upregulate Cd11c in infectious settings). A better gating strategy (migDC, cDC1, cDC2) in the dura and the dCLN should be provided if the authors want to show that those DC are important for antigen transfer in the dCLN. Is there a correlation between DC number in the meninges and parasite load in the brain?

2. The time course of both the WT strain and the attenuated strain loads (gDNA^+^ cysts number) should be given in the brain, meninges, and peripheral tissues over time, to better understand in each model when and where the infection and thus the priming occurs. This should be part of Figure 1, to explain the model.

3. Fig1m is misleading, because there are more DCs around the sinuses in general after infection, not particularly around/in the lymphatics. If the authors were to quantify DC number in Lyve-1 negative area, they would still find an increase. So this does not prove that DCs are going through the lymphatics. As this has already been shown anyway by Kipnis group, they should refer to their study.

4. From Fig1p, it's unclear how the uptake is calculated to be 40% (it looks like 90% to me).

5. In Fig1q, MHC-II stain is everywhere and blurry, so it's hard to see colocalization.

6. It's unclear why the authors looked at DC in the CSF and how they envision the role of CSF DC versus dural DC in priming?

7. In Figure 2, why were 5 hours time point chosen? It seems too fast to have a full picture of migrating DCs (I think in their seminal study, Kipnis used 24h), and too long to only look at soluble antigen drainage.

8. T cell activation is sometimes looked at using IFNγ, or CD44/CD62L strategy, or CD69 (which by the way is not convincing in Figure 3-Supp2). It's unclear why, and this makes the comparison more difficult. It would be interesting to have the total number of CD4 and CD8 also, independently of their activation status.

9. It's convincing that there is a defect in T cell activation in the dCLN upon ligation (Ki67, IFNγ, TNFa which by the way should be in the same figure). Visudyne treatment to locally delete meningeal lymphatics would have been a plus, to prove that this is through dural drainage.

10. In Figure 4, I guess there are no changes neither in CD69+ or CD62L- T cells but have the authors checked? More importantly, what about exhausted and resident CD8^+^ and CD4^+^ T cells?

11. It would have been expected that because of the ligation, T cells would get stuck in the meninges. Have the authors quantified T cell accumulation in the meninges at 6w?

12. In Figure 5, it's interesting to show that even in the chronic phase, T cells are still primed as antigen is still available. What is the contribution of these newly generated OT1 cells in the pool of brain T cells (I guess it should be low, otherwise there would be a phenotype when they are not generated)? And what is the phenotype (activation/exhaustion/residence) of those OT1 generated in the chronic phase when they reach the brain? This would be relevant in the cancer field too.

13. Are CD4^+^ or CD8^+^ T cells the main drivers of parasite clearance (is the brain load higher upon CD4 or CD8 depletion?).

14. The fact that T cells activated in the dCLN do not contribute substantially to the pool of brain T cells in the chronic phase could be expected, as the infection is given i.p. and as other organs seem chronically infected in the periphery. However, it would be interesting to address this question when the pathogen is mainly replicating in the CNS. Could the authors use icv injection of T.g.?

15. It would also be interesting to look at experimental settings in which, during the chronic phase, only the brain is releasing antigen (maybe give a treatment that does not cross the BBB? Or surgically remove/exchange the peripheral organ that is a parasite reservoir). The impact of those chronically primed T cells in the dCLN, if they are the only ones that can be generated, should be very interesting.

Overall, this manuscript convincingly shows that drainage to the dCLN is required to activate dCLN T cells in the chronic phase, but those T cells do not seem to substantially contribute to the pool of brain T cells at this stage. However, some very interesting data could emerge from this model, by looking at the becoming of OT1 generated in the chronic versus acute phase. More importantly, it would be really interesting and impactful to address the role of those dCLN-generated T cells in the acute and chronic phases, in models where the CNS is the main source of antigen, in the acute or chronic phase. These new data would really bring novelty and impact to this study.

*Reviewer #2 (Recommendations for the authors):*

– The gating strategy shown in Figure 1 Suppl1 is for Cd11c+ MHCII+ APCs but does not allow to identify DCs. The MHCII staining looks a bit funky too. There's no real negative population. Can the authors comment on that? Can the authors also show MHCII vs dump?

– In general, there hasn't been used a DC-specific marker combination to identify, characterise and distinguish DCs from other CD11c+ MHCII+ phagocytes (i.e. MdCs or BAMs). The authors should either stick to calling the cells dural APCs throughout the manuscript or perform additional experiments to characterise these cells better and more accurately (for example with an FC panel including CD26 to identify DCs and CD88 to discriminate from MdCs and XCR1 to distinguish cDC1s from cDC2s).

– The authors show an increase in CD11c+ APCs in chronically infected mice, is coincident with chronic infection and increased parasite burden on the brain(?). What happens in the acute stage, e.g. 2 weeks after infection? (Figure 1c)

– Independence of direct infection of the dura: Although, this finding is not central to the main message of the paper it would have been nice if that was supported by a second approach, e.g. microscopy, to exclude the presence of *T. gondii* cysts in the dura mater (Figure 1d)

– In Figures 1k and m there seem to be a lot of Cd11c-YFP cells in naïve mice which can not be seen in Figure 1b. Can the authors comment on that?

– DQ-OVA experiments are nice and convincing but such experiments have been performed previously. Can the authors provide additional information, for instance on the composition of APCs that have sampled Ag in the CNS and enter the lymphatic vessels in the dura mater?

– The CSF data are really intriguing. I was just wondering about the CD80 and Cd86 expression and the FMO data. Was this gated on DCs (Cd11c+ MHCII+) minus CD80 in Figure 1-sup 2, c and Cd11c+ MHCII+ minus CD86 in Figure 1-sup 2, d, respectively? Can the authors show the raw data here? Also, the expression of Cd80 and Cd86 should be shown as MFI and not in frequency since you don't have an isolated population of CD80/86+ cells but rather a shift in the total cells.

– Again, is it possible to distinguish and characterize DQ_OVA+ cells in the DCLNs? Are the CD11cMHCII cDC1s and/or cDC2s? Are there DQ-OVA+ B cells? (Figure 2)

– Is it possible to track OVA specific CD4^+^ T cells as well? (Figure 2) (if not you can transfer of OT-I and OT-II cells for example and characterize their activation – cytokine response and proliferation (CTV dilution), etc.).

– line 208-9 "deep cervical lymph nodes play a distinct role from other lymph nodes in supporting T cells responses …" – That's probably a bit misleading. The data rather suggest/confirm that there are distinct disease stages with the temporal change of the primary site of infection during chronification. The increased T cell responses in the DCLN at later stages just reflect the increased parasite load in the brain. The "role" of the LNs in supporting T cell responses is probably the same though (as stated correctly in 222-24).

– The authors claim that dendritic cells number in the DCLNs are dramatically reduced after blockage of CNS Ag drainage (Figure 3b). As mentioned above the use of Cd11c and MHCII is not sufficient to identify DCs. Are other cells also reduced? B cells for instance? Is there a cell type that is not reduced (negative control)?

– The authors report a decreased activation of DCLNs DCs upon ligation of the lymphatic vessels. However, if they block the influx of activated DCs from the brain it is to be expected that there are fewer CD80/86 expressing cells in the LN (Figure 3c,d). Hence the reduced frequency of CD80/86 expressing cells likely just reflects the reduction in activated APCs rather than a change of the phenotype. Can the authors show the raw data here too?

I would expect that also here it's not possible to gate on CD80/86 positive cells but rather report a change in the MFI. This experiment doesn't really add much information but ok to show in the suppl. data.

– I wouldn't call the decrease of CD69+ T cells in the DCLNs "striking" (line 252). It is rather moderate, particularly for CD8^+^ T cells, which is ok.

– Similar to CD80/86 data for DCs, the phenotypic change of T cells in the DCLNs upon ligation is not surprising as the authors already showed that the influx of activated T cells is reduced. Nevertheless, it confirms the general message that CNS Ag drainage is required for anti-*T. gondii* T cell immunity in the DCLNs. Unfortunately, the IFNγ data do not look very convincing (in particular in Figure 3o). Can the authors include a noninfected control mouse a s well as an unstimulated sample? Maybe also use PMA/ionomycin stimulation as a positive control.

– Can the authors transfer OT-I and OT-II cells and determine the impact of lymphatic vessel ligation on Ag-specific proliferation and cytokine production during infection with the Pru-OVA strain? (please also take the in the brain) (add to figure 3)

– The time point of the ligation seems a bit arbitrary. What's the exact working hypothesis? Some thoughts: The parasite finds its way to the brain despite the presence of a functional T cell response. It is possible that without CNS drainage there would be more parasite load (which is not the case in the present experimental setup). Could the ligation during the infection phase have interfered with the seeding of the brain by itself and "dilute" the effect on parasite load by interfering with T cell responses in the DCLNs? Is it not also possible that reducing Ag drainage would result in a reactivation of the parasite? Can that happen at a later time point? Could the authors block Ag drainage during the chronic phase and look for T cell responses and parasite load at later time points? (e.g. ligation at 6-8 weeks and readout 3 and 6 weeks later?). Is the control of the parasite tachyzoites different from that of bradyzoites? Could the authors check for that (e.g. by qPCR for Bag1 and Sag1 or sth similar or maybe even by histology)?

– The explanation that T cell priming in the periphery is sufficient to lead to T-Gondii specific T cells responses in the CNS is plausible. Does the ligation affect the drainage of CNS Ag to other LNs? Are DC numbers and T cell activation affected by it? The explanation that peripheral Ag is sufficient to induce *T. gondii* T cell which enter the brain to control *T. gondii* seems logical, however, the narrative seemed a bit surprising since the inguinal lymph nodes were introduced as a "negative control" for T cell responses at later time points (during chronic infection). Can the authors show a naïve control of the Nur77GFP experiment (Figure 5 Suppl. 1)

– The discussion seems a bit long and could be a bit more focussed (e.g. remove glymphatic flow part?)

---

## [Author Response]

Essential revision:We all agree that the study warrants publication pending adequate revision.Please find below the key revision points that need to be addressed.1) A better gating strategy to quantify DC subsets with kinetics.

We thank the reviewers for this critical suggestion, and we have addressed this concern by providing a more refined analysis of CD11c^+^ cells using specific markers of type 1 and type 2 conventional dendritic cells (cDC1s and cDC2s, respectively). We developed a gating strategy using a framework defined by Guilliams et al. (2016) and consistent with well-established dendritic cell lineage markers (Merad, Sathe et al. 2013). This gating strategy is provided in Figure 1 Supp 1. Briefly, we used a lineage dump gate to exclude T cells (CD3^+^), B cells (CD19^+^), NK cells (NK1.1^+^), neutrophils (Ly6G^+^), and erythroid cells (TER-119^+^) from our CD45^+^ population of interest. We then gated on the CD64^-^ subset of CD11c^hi^MHCII^hi^ cells to exclude macrophages (CD64^+^) that upregulated CD11c in the setting of inflammation. Next, we gated on CD26^+^ cells, which defines FLT3-dependent conventional dendritic cells (Nakano, Moran et al. 2015). Finally, we gated on XCR1^+^SIRPα^-^ cDC1s and XCR1^-^SIRPα^+^ cDC2s, markers that reliably track with expression of the cDC1 and cDC2 lineage-defining transcription factors IRF8 and IRF4, respectively (Guilliams, Dutertre et al. 2016). This gating strategy was employed for analysis of dendritic cells in the dural meninges (Figure 1f-k, Figure 1 Supp 2) and for analysis of DQ-OVA^+^ dendritic cells in the DCLNs (Figure 2 Supp 1).

To understand the kinetics of dendritic cell accumulation in the dural meninges, we quantified cDC1 and cDC2 number and frequency in naïve, acutely infected (2 wpi), and chronically infected (6 wpi) mice (Figure 1f-g, Figure 1 Supp 2a-c). We additionally examined the kinetics of dendritic cell activation by measuring cDC1 and cDC2 expression of CD80, CD86, and MHC class II (Figure 1i-k, Figure 1 Supp 2e-g). Interestingly, we did not observe a difference in cDC1 or cDC2 number or frequency between naïve and acutely infected mice. However, in chronically infected mice, there was a striking increase in cDC1 and cDC2 number and a shift in frequency of the two populations. Additionally, expression of co-stimulatory molecules peaked at 6 wpi. We also measured parasite burden in the brain and dural meninges at these same time points in order to determine whether there was a relationship between dendritic cell accumulation/activation and parasite burden in either tissue (Figure 1h). In fact, expansion and activation of the cDC1 and cDC2 populations in the dural meninges corresponded to increased parasite burden in the brain at 6 wpi and not to parasite burden in the dural meninges. Indeed, the dural meninges lacked evidence of direct infection at 6 wpi both by real-time PCR and by fluorescent immunohistochemistry (Figure 1h, Figure 1 Supp 2d).

In order to understand the functional capacity of cDC1s and cDC2s both in the dural meninges and in the draining deep cervical lymph nodes (DCLNs), we quantified the frequency of cDC1s and cDC2s at 6 wpi that harbored processed DQ-OVA either 2 h or 24 h after injection into the CSF (Figure 1 Supp 2h-j, Figure 2 Supp 1). We chose these timepoints because soluble CSF-borne protein can be detected in the DCLNs within 2 h of i.c.m. injection in infected mice (Figure 3 Supp 1b) and previous experiments have revealed that dendritic cells from the dural meninges traffic to the DCLNs by 24 h (Louveau, Herz et al. 2018). Our studies demonstrated that cDC1s and cDC2s were equally capable of capturing CSF-derived protein during chronic brain infection.

2) A better definition of T cell responses with kinetics and fate of OT1/ dCLN generated T cell generated in the chronic versus acute phase.

In order to characterize the phenotype of OT-Is generated in the DCLN and OT-Is that trafficked to the brain, we designed an experiment in which OT-I cells (CD45.1 congenic) were adoptively transferred to C57BL/6 mice either during acute infection (at 10 days post-infection) or during chronic infection (at 5 weeks post-infection) (Figure 2 Supp 3a). Seven days later, we measured expression of relevant markers in the DCLNs and brain, including PD-1, TIM-3, and LAG-3, and markers that have been shown to define effector and memory-like T cells during *T. gondii* infection, including Trm-like cells (Chu, Chan et al. 2016, Landrith, Sureshchandra et al. 2017). We examined the phenotype of both the transferred CD45.1^+^ T cell population and the endogenous CD45.1^-^ SIINFEKL-specific CD8^+^ T cell population. Briefly, we observed low expression of co-inhibitory molecules on both the endogenous and transferred populations and, as expected, saw an increase in the memory subsets of endogenous parasite-specific T cells during chronic infection. The results of these experiments are shown in Figure 2 Supp 3 and Figure 2 Supp 4. We also examined expression of these markers on the polyclonal CD4^+^ and CD8^+^ T cell populations in sham or ligated mice, as shown in Figure 4 Supp 2, Figure 5 Supp 2, and Figure 5 Supp 3. From these experiments, we can conclude that there is a reduction in expression of co-inhibitory molecules in the DCLNs after ligation, consistent with a reduction in T cell activation, and no differences in co-inhibitory molecule expression or memory status in the brain after ligation.

3) Kinetics of parasite replication in the periphery (e.g. in tissue draining to ILN) and in the dural meninges, for WT and/or Pru-OVA strains to provide a better context to understand the localization of T cell priming in chronic and acute phases.

We agree with the reviewers that this is an essential experiment. Therefore, following infection with Pru-OVA, we measured parasite burden by real-time PCR in several CNS and peripheral compartments at 7 days post-infection (dpi), 2 wpi, 3 wpi, and 6 wpi. These data are provided in Figure 2f, in order to provide context to the analysis of parasite-specific T cell responses in the deep cervical lymph nodes (DCLN) and inguinal lymph nodes (ILN) over the course of acute and chronic infection. Parasite burden was measured in the brain and dural meninges, as these sites are drained by the DCLNs, and in the subcutaneous tissue underlying the skin of the flank, as this site drains to the ILNs. Parasite burden was also measured in the DCLNs and ILNs themselves, which may be direct sites of infection themselves. Finally, parasite burden was measured in the quadriceps femoris skeletal muscle, as it has been reported that *T. gondii* persists in skeletal muscle during chronic infection, although detection by real-time PCR has demonstrated levels to be low or undetectable (Jin, Blair et al. 2017, Melchor, Hatter et al. 2020).

We find that parasite burden is highest, among the tissues studied, in the inguinal lymph nodes during acute infection (7 dpi) and undetectable in the brain and dural meninges. By contrast, as infection progressed into the chronic stage, parasite burden increased in the brain and by 6 wpi was detectable only at this site. These results suggest that during acute infection, parasite (and *T. gondii* antigen) is greatest in lymph nodes that drain peripheral tissues, such as the ILNs. The high levels of parasite gDNA in the ILNs likely reflects drainage of parasite or infected cells from infected peripheral tissues or direct infection of the lymph nodes themselves. During chronic infection, parasite could be detected only in the brain, and not in the dural meninges or skeletal muscle. These data explain the robust T cell responses in the DCLNs observed at 6 wpi (Figure 2c-e) and suggest that the antigen-dependent T cell responses observed in the ILNs during chronic infection (Figure 6) may result from residual antigen in the ILNs persisting after the resolution of acute infection.

Reviewer #1 (Recommendations for the authors):1. In Figure 1, DC are quantified using CD11cYFP reporter, or CD11c gating strategy (which by the way is not very clean in several figures), which is not sufficient to prove that those are DCs (for instance, activated macrophages can upregulate Cd11c in infectious settings). A better gating strategy (migDC, cDC1, cDC2) in the dura and the dCLN should be provided if the authors want to show that those DC are important for antigen transfer in the dCLN. Is there a correlation between DC number in the meninges and parasite load in the brain?

The reviewer brings up an important point, and we have addressed this concern by providing a more refined analysis of CD11c^+^ cells by spectral flow cytometry, using very specific markers of cDC1s and cDC2s. Our new gating strategy is discussed above (Editor’s Essential Revisions, Point 1). Like the reviewer, we were also interested in the question of whether DC number in the meninges tracked with parasite load in the brain or meninges. For this reason, we measured cDC1 and cDC2 number in naïve, acutely infected (2 wpi), and chronically infected (6 wpi) mice (Figure 1f-g) and, in a separate cohort of mice, measured parasite burden by real-time PCR in the brain and dural meninges at these same time points (Figure 1h). Interestingly, cDC1 and cDC2 accumulation in the dural meninges (as well as activation of cDC1s and cDC2s, as measured by CD80, CD86, and MHC class II expression, Figure 1i-k, Figure 1 Supp 2e-g) corresponded to increased parasite burden in the brain and not the dural meninges. In fact, parasite gDNA was not detected in the dural meninges of any mice at 6 wpi and could only be detected in 4/7 mice at 2 wpi (Figure 1h).

2. The time course of both the WT strain and the attenuated strain loads (gDNA^+^ cysts number) should be given in the brain, meninges, and peripheral tissues over time, to better understand in each model when and where the infection and thus the priming occurs. This should be part of Figure 1, to explain the model.

We thank the reviewer for this important suggestion. We have addressed the point experimentally and report our findings in Figure 2f, as described above (Editor’s Essential Revisions, Point 3).

3. Fig1m is misleading, because there are more DCs around the sinuses in general after infection, not particularly around/in the lymphatics. If the authors were to quantify DC number in Lyve-1 negative area, they would still find an increase. So this does not prove that DCs are going through the lymphatics. As this has already been shown anyway by Kipnis group, they should refer to their study.

We concede that there is an increase in the CD11c^+^ number in both the LYVE1^+^ area and the LYVE1^-^ area and that this makes it challenging to conclude that dendritic cells trafficking through the vessels is increased with infection. We have clarified this point in the text and referred to the Kipnis study that already demonstrated dendritic cell trafficking through these vessels to the deep cervical lymph nodes (Louveau, Herz et al. 2018).

4. From Fig1p, it's unclear how the uptake is calculated to be 40% (it looks like 90% to me).

We apologize for this inconsistency. In order to achieve a more precise picture of how cDC1s and cDC2s participate in CSF-borne antigen capture, this experiment was completed again using our new gating strategy for dendritic cells. These data are now presented in Figure 1 Supp 2h-j, and we have aimed to provide clearer and more representative flow plots showing the DQ-OVA^+^ and DQ-OVA^-^ dendritic cell populations, consistent with the quantification shown in Figure 1 Supp 2j.

5. In Fig1q, MHC-II stain is everywhere and blurry, so it's hard to see colocalization.

We thank the reviewer for identifying this difficulty in interpretation and we agree. Therefore, we have a provided an alternative representative image (Figure 1m) that we believe is clearer.

6. It's unclear why the authors looked at DC in the CSF and how they envision the role of CSF DC versus dural DC in priming?

The reviewer’s point is well taken. We have changed our language in the text to make the rationale for this experiment more clear. In short, seminal studies by the Kipnis lab demonstrated that dendritic cells injected into the CSF exit the CNS through meningeal lymphatic vessels (Louveau, Herz et al. 2018). Because we observed an increase in the number of CD11c+ cells in the area of the meninges around the lymphatic vessels, we sought to determine whether dendritic cells were present in the CSF. Though this experiment does not directly address the different functions of CSF and dural APCs, we believe our findings represent a valuable first step towards understanding the distinct functions of APCs in the different compartments of the CNS and CNS border tissues.

7. In Figure 2, why were 5 hours time point chosen? It seems too fast to have a full picture of migrating DCs (I think in their seminal study, Kipnis used 24h), and too long to only look at soluble antigen drainage.

The 5-hour time point was selected based on a pilot experiment used to identify when emission of digested DQ-OVA products becomes detectable in the dural meninges. We agree with the reviewer’s criticism that this time point limits our ability to discriminate between drainage of soluble antigen and transport of antigen by migratory DCs, although the gold standard would be to design an experiment using photoconvertible dendritic cells, as was already performed by Louveau et al. in naïve animals (2018). Nonetheless, in order to achieve a preliminary understanding of whether DQ-OVA emission in the DCLNs is dependent on drainage of soluble antigen or transport via migratory DCs, we performed a new set of experiments examining the presence of CSF-derived DQ-OVA within cDC1s and cDC2s at 2 h and 24 h after i.c.m. injection (Figure 1 Supp 2h-j, Figure 2 Supp 1). However, because the frequency of DQ-OVA^+^ cDC1s and cDC2s decreased from 30% at 2 h to 5% at 24 h in the dural meninges but did not change in the DCLNs, it remains unclear whether a large portion of DQ-OVA^+^ cells migrated from the dura or if the fluorescent signal produced by DQ-OVA cleavage products declines over time with additional processing.

8. T cell activation is sometimes looked at using IFNγ, or CD44/CD62L strategy, or CD69 (which by the way is not convincing in Figure 3-Supp2). It's unclear why, and this makes the comparison more difficult. It would be interesting to have the total number of CD4 and CD8 also, independently of their activation status.

We apologize that the rationale for these parameters was not clear. The decision to assess activation status of T cells using CD44^hi^CD62L^lo^ or CD69 was based on the kinetics of upregulation of these molecules on activated T cells. Because antigen-experienced CD44^hi^CD62L^lo^ T cells continue to exhibit this phenotype after activation, measuring the total number of CD44^hi^CD62L^lo^ T cells provided insight into how the overall magnitude of the T cell response evolves in the DCLNs and ILNs over the course of infection. For the ligation studies, however, we were interested in identifying changes in activation that occurred in a short window of time (in the three week interval of ligation), and measuring CD69 proved to be a more sensitive read-out for this because CD69 is upregulated only on the most recently activated T cells (Simms and Ellis 1996). The reason for examining proliferation and cytokine expression was to understand how restriction of meningeal lymphatic outflow affected not just markers of T cell activation but also functional read-outs of T cell activation and T cell effector capacity. As suggested, we included the total number of CD4^+^ and CD8^+^ T cells in the DCLNs after ligation (Figure 4 Supp 1a-b).

9. It's convincing that there is a defect in T cell activation in the dCLN upon ligation (Ki67, IFNγ, TNFa which by the way should be in the same figure). Visudyne treatment to locally delete meningeal lymphatics would have been a plus, to prove that this is through dural drainage.

We agree that Visudyne would have been an outstanding complementary approach for demonstrating dependence of T cell responses on meningeal lymphatic vessels. Unfortunately, we did not have access to the instrumentation required for completing this experiment.

10. In Figure 4, I guess there are no changes neither in CD69+ or CD62L- T cells but have the authors checked? More importantly, what about exhausted and resident CD8^+^ and CD4^+^ T cells?

As other groups have reported (Reichmann, Villegas et al. 1999, Wilson, Harris et al. 2009), we find that almost the entire population of CD4^+^ and CD8^+^ T cells entering the brain are CD44^hi^CD62L^lo^. Following ligation, we did not observe differences in the activation status of T cells in the brain, and we have included these data in Figure 5 Supp 1. We also examined markers of T cell exhaustion (PD-1, LAG-3, and TIM-3) and report decreased expression of these molecules in the DCLNs (Figure 4 Supp 2) and no differences in expression of these molecules in the brain (Figure 5 Supp 2) following ligation. We examined CD69^+^CD103^+^ Trm-like cells (Landrith, Sureshchandra et al. 2017) and report no differences in the brain after ligation (Figure 5 Supp 3).

11. It would have been expected that because of the ligation, T cells would get stuck in the meninges. Have the authors quantified T cell accumulation in the meninges at 6w?

We examined CD4^+^ and CD8^+^ T cell numbers in the meninges and observed no differences after ligation. These data may reflect alternative outflow routes, such as the superficial cervical lymph nodes, or lack of survival of retained T cells.

**Author response image 1. sa2fig1:** 

12. In Figure 5, it's interesting to show that even in the chronic phase, T cells are still primed as antigen is still available. What is the contribution of these newly generated OT1 cells in the pool of brain T cells (I guess it should be low, otherwise there would be a phenotype when they are not generated)? And what is the phenotype (activation/exhaustion/residence) of those OT1 generated in the chronic phase when they reach the brain? This would be relevant in the cancer field too.

In order to characterize the phenotype of OT-Is generated in the DCLN and OT-Is that trafficked to the brain, we designed an experiment in which OT-I cells (CD45.1 congenic) were adoptively transferred to C57BL/6 mice either during acute infection (at 10 days post-infection) or during chronic infection (at 5 weeks post-infection) (Figure 2 Supp 3a). Seven days later, we measured expression of relevant markers in the DCLNs and brain, including PD-1, TIM-3, and LAG-3, and markers that have been shown to define effector and memory-like T cells during *T. gondii* infection, including Trm-like cells (Chu, Chan et al. 2016, Landrith, Sureshchandra et al. 2017). The results of these experiments are shown in Figure 2 Supp 3 and Figure 2 Supp 4 and described above (Editor’s Essential Revisions, Point 2).

13. Are CD4^+^ or CD8^+^ T cells the main drivers of parasite clearance (is the brain load higher upon CD4 or CD8 depletion?).

CD4^+^ and CD8^+^ T cells are both essential for parasite clearance and host survival during the chronic stage of infection. Studies performed by the Sher group demonstrated that combined depletion of CD4^+^ and CD8^+^ T cells leads to rapid mortality of chronically infected mice and depletion of either CD4^+^ or CD8^+^ T cells leads to increased parasite burden in the brain (Gazzinelli, Xu et al. 1992).

14. The fact that T cells activated in the dCLN do not contribute substantially to the pool of brain T cells in the chronic phase could be expected, as the infection is given i.p. and as other organs seem chronically infected in the periphery. However, it would be interesting to address this question when the pathogen is mainly replicating in the CNS. Could the authors use icv injection of T.g.?

We agree with the reviewer that confining infection to the brain would be an excellent proof-of-concept experiment to determine whether CNS lymphatic drainage to the DCLNs becomes more important when peripheral sources of antigen are minimized. We attempted this experiment by directly injected parasite into the cortex. However, in our pilot study, the mice reached their humane endpoint within 9 days, and by this time, only one of three mice had mounted a parasite-specific T cell response in the brain.

15. It would also be interesting to look at experimental settings in which, during the chronic phase, only the brain is releasing antigen (maybe give a treatment that does not cross the BBB? Or surgically remove/exchange the peripheral organ that is a parasite reservoir). The impact of those chronically primed T cells in the dCLN, if they are the only ones that can be generated, should be very interesting.

We agree with the reviewer that this would be a valuable experiment. Although the transgenic strain of parasite needed to make this experiment feasible is being developed by other groups—one that expresses OVA under the control of the bradyzoite-specific BAG1 promoter—this tool is unfortunately not yet available.

Overall, this manuscript convincingly shows that drainage to the dCLN is required to activate dCLN T cells in the chronic phase, but those T cells do not seem to substantially contribute to the pool of brain T cells at this stage. However, some very interesting data could emerge from this model, by looking at the becoming of OT1 generated in the chronic versus acute phase. More importantly, it would be really interesting and impactful to address the role of those dCLN-generated T cells in the acute and chronic phases, in models where the CNS is the main source of antigen, in the acute or chronic phase. These new data would really bring novelty and impact to this study.Reviewer #2 (Recommendations for the authors):– The gating strategy shown in Figure 1 Suppl1 is for Cd11c+ MHCII+ APCs but does not allow to identify DCs. The MHCII staining looks a bit funky too. There's no real negative population. Can the authors comment on that? Can the authors also show MHCII vs dump?

We agree with the reviewer that our gating strategy for antigen-presenting cells was not sufficient to accurately identify the population of CD11c^hi^MHC II^hi^ cells as dendritic cells. For this reason, we completed a new analysis of dendritic cells in the dural meninges and DCLNs using a more precise gating strategy, as described above in the Editor’s Essential Revisions, Point 1. We also provide in Author response image 3 a representative contour plot showing MHC II v Lineage (dump). It is possible that the MHC II staining appears higher in the infected dura because the CD45+ population is dominated by myeloid cells that have upregulated MHC II in response to infection.

**Author response image 3. sa2fig3:** 

– In general, there hasn't been used a DC-specific marker combination to identify, characterise and distinguish DCs from other CD11c+ MHCII+ phagocytes (i.e. MdCs or BAMs). The authors should either stick to calling the cells dural APCs throughout the manuscript or perform additional experiments to characterise these cells better and more accurately (for example with an FC panel including CD26 to identify DCs and CD88 to discriminate from MdCs and XCR1 to distinguish cDC1s from cDC2s).

We agree with the reviewer and have provided a description of our new gating strategy above, in the Editor’s Essential Revisions, Point 1.

– The authors show an increase in CD11c+ APCs in chronically infected mice, is coincident with chronic infection and increased parasite burden on the brain(?). What happens in the acute stage, e.g. 2 weeks after infection? (Figure 1c)

As described above (see Reviewer 1, Point 1 and Editor’s Essential Revisions, Point 1), we observed an increased accumulation of cDC1s and cDC2s in the dural meninges during chronic infection, and this increase in number corresponded to increased parasite burden in the brain but not the meninges (Figure 1f-h). There is no difference in cDC1 or cDC2 number in the dural meninges of naive and acutely infected mice (2 wpi). With the exception of a mild increase in CD80 expression by cDC2s during acute infection, upregulation of co-stimulatory molecules and MHC class II occurred during chronic infection (6 wpi) (Figure 1i-k, Figure 1 Supp 2e-g).

– Independence of direct infection of the dura: Although, this finding is not central to the main message of the paper it would have been nice if that was supported by a second approach, e.g. microscopy, to exclude the presence of *T. gondii* cysts in the dura mater (Figure 1d)

We appreciate the reviewer’s suggestion and have provided representative images demonstrating the absence of parasite in the dural meninges during chronic infection using a polyclonal α-ME49 antibody (Figure 1 Supp 2d).

– In Figures 1k and m there seem to be a lot of Cd11c-YFP cells in naïve mice which can not be seen in Figure 1b. Can the authors comment on that?

We apologize for this confusion. The apparent discrepancy resulted from poor downsampling of the tilescan when being prepared for the PDF file. In order to resolve this issue, we enlarged the tilescan and increased the pixel density to 450 ppi (Figure 1a).

– DQ-OVA experiments are nice and convincing but such experiments have been performed previously. Can the authors provide additional information, for instance on the composition of APCs that have sampled Ag in the CNS and enter the lymphatic vessels in the dura mater?

We appreciate the reviewer’s suggestion and have followed up our initial DQ-OVA studies with new experiments examining the presence of DQ-OVA in cDC1s and cDC2s (Figure 1 Supp 2h-j, Figure 2 Supp 1). These experiments revealed that cDC1s and cDC2s were equally capable of capturing CSF-derived protein during chronic brain infection and showed that the frequency of DQ-OVA^+^ cells remained equivalent in the DCLNs at 2 h and 24 h post-injection, despite falling from 30% to 5% in the dural meninges for both cDC1s and cDC2s.

– The CSF data are really intriguing. I was just wondering about the CD80 and Cd86 expression and the FMO data. Was this gated on DCs (Cd11c+ MHCII+) minus CD80 in Figure 1-sup 2, c and Cd11c+ MHCII+ minus CD86 in Figure 1-sup 2, d, respectively? Can the authors show the raw data here? Also, the expression of Cd80 and Cd86 should be shown as MFI and not in frequency since you don't have an isolated population of CD80/86+ cells but rather a shift in the total cells.

We apologize for the confusion regarding the gating strategy and quantification. Indeed, we measured frequency of CD80^+^ and CD86^+^ CD11c^hi^MHC II^hi^ cells isolated from the CSF and the positive populations were determined using FMOs for CD80 and CD86. We elected to examine frequency rather than MFI in order to gauge what proportion of the overall APC population expressed any level of these co-stimulatory molecules. Lacking a comparison group (APCs were absent in the CSF of naïve mice), we believed this read-out would be more informative than MFI alone. The raw data, including the FMOs used for gating and the quantification, are now shown in Figure 1 Supp 3c-e.

– Again, is it possible to distinguish and characterize DQ_OVA+ cells in the DCLNs? Are the CD11cMHCII cDC1s and/or cDC2s? Are there DQ-OVA+ B cells? (Figure 2)

We appreciate the reviewer’s suggestion and have provided a more detailed analysis of the APCs that harbor DQ-OVA in the DCLNs, as explained above in Point 6. The frequencies of cDC1s and cDC2s in the DCLNs that harbored DQ-OVA digestion products 2 or 24 h after i.c.m. injection were equivalent, as shown in Figure 2 Supp 1.

– Is it possible to track OVA specific CD4^+^ T cells as well? (Figure 2) (if not you can transfer of OT-I and OT-II cells for example and characterize their activation – cytokine response and proliferation (CTV dilution), etc.).

Unfortunately, OVA-specific CD4^+^ T cells or OT-II cells cannot be readily tracked in our C57BL/6 mouse model of infection. It is for this reason that most groups studying *T. gondii* infection in C57BL/6 mice track parasite-specific T cell responses using the MHC I-SIINFEKL tetramer and the OT-I transfer system (Schaeffer, Han et al. 2009, Wilson, Harris et al. 2009).

– line 208-9 "deep cervical lymph nodes play a distinct role from other lymph nodes in supporting T cells responses …" – That's probably a bit misleading. The data rather suggest/confirm that there are distinct disease stages with the temporal change of the primary site of infection during chronification. The increased T cell responses in the DCLN at later stages just reflect the increased parasite load in the brain. The "role" of the LNs in supporting T cell responses is probably the same though (as stated correctly in 222-24).

We completely agree with the reviewer and have removed this language from the text. We conclude, instead, that “[t]hese results likely reflect the distinct sources of antigen that each lymph node site drains during the different stages of infection.”

– The authors claim that dendritic cells number in the DCLNs are dramatically reduced after blockage of CNS Ag drainage (Figure 3b). As mentioned above the use of Cd11c and MHCII is not sufficient to identify DCs. Are other cells also reduced? B cells for instance? Is there a cell type that is not reduced (negative control)?

We performed the experiment again using our refined gating strategy for cDC1s and cDC2s and interestingly observed no difference in the total number of cDC1s or cDC2s in the DCLNs after ligation (Figure 3b-c). However, we did observe a significant reduction in the frequency and number of CD103+ cDC1s and a trending decrease in the frequency and number of CD103+ cDC2s after ligation (Figure 3d-i). These results are consistent with the CD103+ subset representing a migratory dendritic cell population that trafficked from the dural meninges.

– The authors report a decreased activation of DCLNs DCs upon ligation of the lymphatic vessels. However, if they block the influx of activated DCs from the brain it is to be expected that there are fewer CD80/86 expressing cells in the LN (Figure 3c,d). Hence the reduced frequency of CD80/86 expressing cells likely just reflects the reduction in activated APCs rather than a change of the phenotype. Can the authors show the raw data here too?I would expect that also here it's not possible to gate on CD80/86 positive cells but rather report a change in the MFI. This experiment doesn't really add much information but ok to show in the suppl. data.

After performing this experiment using our refined gating strategy for cDC1s and cDC2s, we found that even though there is no difference in the total number of cDC1s and cDC2s in the DCLNs, the expression of co-stimulatory molecules by these cells is significantly reduced (Figure 3j-q). This difference may be the result of reduced trafficking of activated dendritic cells to the DCLNs, but it also raises the question as to whether pro-inflammatory factors draining from the brain, CSF, or meninges contribute to the activation of lymphoid-resident DCs. As the reviewer suggests, we have provided the raw data and reported expression as MFI.

– I wouldn't call the decrease of CD69+ T cells in the DCLNs "striking" (line 252). It is rather moderate, particularly for CD8^+^ T cells, which is ok.

Per the reviewer’s suggestion, we have changed the language of the text in order to avoid the misleading characterization of the change as “striking”.

– Similar to CD80/86 data for DCs, the phenotypic change of T cells in the DCLNs upon ligation is not surprising as the authors already showed that the influx of activated T cells is reduced. Nevertheless, it confirms the general message that CNS Ag drainage is required for anti-*T. gondii* T cell immunity in the DCLNs. Unfortunately, the IFNγ data do not look very convincing (in particular in Figure 3o). Can the authors include a noninfected control mouse a s well as an unstimulated sample? Maybe also use PMA/ionomycin stimulation as a positive control.

We appreciate the reviewer’s concerns, in particular given the low degree of IFN-γ expression by CD8^+^ T cells. We have thus provided unstimulated CD4^+^ and CD8^+^ T cells as negative controls (“No STAg”), which should help with the interpretation of these data (Figure 4j, m, p).

– Can the authors transfer OT-I and OT-II cells and determine the impact of lymphatic vessel ligation on Ag-specific proliferation and cytokine production during infection with the Pru-OVA strain? (please also take the in the brain) (add to figure 3)

We performed this experiment and observed a trending decrease in the total number of OT-Is in the DCLNs after ligation but no difference in the total number of OT-Is in the brain (Author response image 4, d). Additionally, there was no observed difference in CellTrace Violet dilution at either site (Author response image 4, e-f). The lack of a difference in dye dilution among OT-I cells in the DCLNs contrasts with the reduction in Ki67 expression that we observed in the polyclonal CD8^+^ T cell population after ligation (Figure 4g-h). This disparity may reflect the self-selecting nature of proliferative (CTV+) cells, which would become overrepresented in the OT-I population as OT-I cells that fail to proliferate die off.

**Author response image 4. sa2fig4:** 

– The time point of the ligation seems a bit arbitrary. What's the exact working hypothesis? Some thoughts: The parasite finds its way to the brain despite the presence of a functional T cell response. It is possible that without CNS drainage there would be more parasite load (which is not the case in the present experimental setup). Could the ligation during the infection phase have interfered with the seeding of the brain by itself and "dilute" the effect on parasite load by interfering with T cell responses in the DCLNs? Is it not also possible that reducing Ag drainage would result in a reactivation of the parasite? Can that happen at a later time point? Could the authors block Ag drainage during the chronic phase and look for T cell responses and parasite load at later time points? (e.g. ligation at 6-8 weeks and readout 3 and 6 weeks later?). Is the control of the parasite tachyzoites different from that of bradyzoites? Could the authors check for that (e.g. by qPCR for Bag1 and Sag1 or sth similar or maybe even by histology)?

One of the limitations of the ligation approach is the possibility that collateral vessels might reroute lymph flow around the ligated vessels and reduce the length of time for which our approach to blocking meningeal lymphatic outflow remains efficacious (Blum, Proulx et al. 2013, Kwon, Agollah et al. 2014). We demonstrated that ligation effectively blocks CSF outflow to the DCLNs for three weeks (Figure 3 Supp 1a-b). With this interval in mind, we decided to perform ligation at three weeks post-infection when T cells begin to display a response against the parasite in the DCLNs (Figure 2), a point we have made more clear in the text. We do not believe that ligation would have a major impact on parasite seeding of the brain because parasite burden in the blood peaks between 7-10 dpi and parasite has been cleared from peripheral tissues by 3 wpi (Saeij, Boyle et al. 2005, Djurkovic-Djakovic, Djokic et al. 2012). To assess whether ligation affected control of tachyzoites or bradyzoites differentially, we measured total parasite gDNA by real-time PCR (Figure 5f) and cyst count by microscopy (Figure 5 Supp 1e). Because there was no difference in total parasite gDNA or cyst number, it is unlikely that there is a difference in tachyzoite or bradyzoite burden.

– The explanation that T cell priming in the periphery is sufficient to lead to *T-Gondii* specific T cells responses in the CNS is plausible. Does the ligation affect the drainage of CNS Ag to other LNs? Are DC numbers and T cell activation affected by it? The explanation that peripheral Ag is sufficient to induce T. gondii T cell which enter the brain to control T. gondii seems logical, however, the narrative seemed a bit surprising since the inguinal lymph nodes were introduced as a "negative control" for T cell responses at later time points (during chronic infection). Can the authors show a naïve control of the Nur77GFP experiment (Figure 5 Suppl. 1)

The reviewer brings up an excellent point. The DCLNs are not the only drainage site for CSF. Other groups have shown that the superficial cervical lymph nodes (SCLNs) also drain components of the CSF (Ma, Ineichen et al. 2017). As suggested, we tested whether ligation of the lymphatic vessels afferent to the DCLNs affects drainage of CSF-derived antigen to the SCLNs or T cell responses in these nodes. Indeed, we observed no change in outflow (Figure 3 Supp 1c) and no change in CD4^+^ or CD8^+^ T cell activation (Figure 4 Supp 1g-l). Because the SCLNs represent another outflow site, we also tested whether blocking drainage to both the DCLNs and SCLNs would have an impact on the T cell response in the brain (Figure 5 Supp 4). However, there was no difference in CD4^+^ or CD8^+^ T cell number in the brain (Figure 5 Supp 4c-d), providing additional support to the conclusion that meningeal lymphatic drainage is dispensable for T cell responses in the brain, and that T cell responses at peripheral sites are likely playing an important role in sustaining T cell responses in the brain.

– The discussion seems a bit long and could be a bit more focussed (e.g. remove glymphatic flow part?)

We agree with the reviewer’s comment and have removed the discussion of glymphatic flow to be more focused.

References

Blum, K. S., S. T. Proulx, P. Luciani, J. C. Leroux and M. Detmar (2013). "Dynamics of lymphatic regeneration and flow patterns after lymph node dissection." Breast Cancer Res Treat 139(1): 81-86.

Chu, H. H., S. W. Chan, J. P. Gosling, N. Blanchard, A. Tsitsiklis, G. Lythe, N. Shastri, C. Molina-Paris and E. A. Robey (2016). "Continuous Effector CD8(+) T Cell Production in a Controlled Persistent Infection Is Sustained by a Proliferative Intermediate Population." Immunity 45(1): 159-171.

Djurkovic-Djakovic, O., V. Djokic, M. Vujanic, T. Zivkovic, B. Bobic, A. Nikolic, K. Slavic, I. Klun and V. Ivovic (2012). "Kinetics of parasite burdens in blood and tissues during murine toxoplasmosis." Exp Parasitol 131(3): 372-376.

Gazzinelli, R., Y. Xu, S. Hieny, A. Cheever and A. Sher (1992). "Simultaneous depletion of CD4^+^ and CD8^+^ T lymphocytes is required to reactivate chronic infection with *Toxoplasma gondii*." J Immunol 149(1): 175-180.

Guilliams, M., C. A. Dutertre, C. L. Scott, N. McGovern, D. Sichien, S. Chakarov, S. Van Gassen, J. Chen, M. Poidinger, S. De Prijck, S. J. Tavernier, I. Low, S. E. Irac, C. N. Mattar, H. R. Sumatoh, G. H. L. Low, T. J. K. Chung, D. K. H. Chan, K. K. Tan, T. L. K. Hon, E. Fossum, B. Bogen, M. Choolani, J. K. Y. Chan, A. Larbi, H. Luche, S. Henri, Y. Saeys, E. W. Newell, B. N. Lambrecht, B. Malissen and F. Ginhoux (2016). "Unsupervised High-Dimensional Analysis Aligns Dendritic Cells across Tissues and Species." Immunity 45(3): 669-684.

Jin, R. M., S. J. Blair, J. Warunek, R. R. Heffner, I. J. Blader and E. A. Wohlfert (2017). "Regulatory T Cells Promote Myositis and Muscle Damage in *Toxoplasma gondii* Infection." J Immunol 198(1): 352-362.

Kwon, S., G. D. Agollah, G. Wu and E. M. Sevick-Muraca (2014). "Spatio-temporal changes of lymphatic contractility and drainage patterns following lymphadenectomy in mice." PLoS One 9(8): e106034.

Landrith, T. A., S. Sureshchandra, A. Rivera, J. C. Jang, M. Rais, M. G. Nair, I. Messaoudi and E. H. Wilson (2017). "CD103(+) CD8 T Cells in the Toxoplasma-Infected Brain Exhibit a Tissue-Resident Memory Transcriptional Profile." Front Immunol 8: 335.

Louveau, A., J. Herz, M. N. Alme, A. F. Salvador, M. Q. Dong, K. E. Viar, S. G. Herod, J. Knopp, J. C. Setliff, A. L. Lupi, S. Da Mesquita, E. L. Frost, A. Gaultier, T. H. Harris, R. Cao, S. Hu, J. R. Lukens, I. Smirnov, C. C. Overall, G. Oliver and J. Kipnis (2018). "CNS lymphatic drainage and neuroinflammation are regulated by meningeal lymphatic vasculature." Nat Neurosci 21(10): 1380-1391.

Ma, Q., B. V. Ineichen, M. Detmar and S. T. Proulx (2017). "Outflow of cerebrospinal fluid is predominantly through lymphatic vessels and is reduced in aged mice." Nat Commun 8(1): 1434.

Melchor, S. J., J. A. Hatter, E. A. L. Castillo, C. M. Saunders, K. A. Byrnes, I. Sanders, D. Abebayehu, T. H. Barker and S. E. Ewald (2020). "*T. gondii* infection induces IL-1R dependent chronic cachexia and perivascular fibrosis in the liver and skeletal muscle." Sci Rep 10(1): 15724.

Merad, M., P. Sathe, J. Helft, J. Miller and A. Mortha (2013). "The dendritic cell lineage: ontogeny and function of dendritic cells and their subsets in the steady state and the inflamed setting." Annu Rev Immunol 31: 563-604.

Nakano, H., T. P. Moran, K. Nakano, K. E. Gerrish, C. D. Bortner and D. N. Cook (2015). "Complement receptor C5aR1/CD88 and dipeptidyl peptidase-4/CD26 define distinct hematopoietic lineages of dendritic cells." J Immunol 194(8): 3808-3819.

Reichmann, G., E. N. Villegas, L. Craig, R. Peach and C. A. Hunter (1999). "The CD28/B7 interaction is not required for resistance to *Toxoplasma gondii* in the brain but contributes to the development of immunopathology." J Immunol 163(6): 3354-3362.

Saeij, J. P., J. P. Boyle, M. E. Grigg, G. Arrizabalaga and J. C. Boothroyd (2005). "Bioluminescence imaging of *Toxoplasma gondii* infection in living mice reveals dramatic differences between strains." Infect Immun 73(2): 695-702.

Schaeffer, M., S. J. Han, T. Chtanova, G. G. van Dooren, P. Herzmark, Y. Chen, B. Roysam, B. Striepen and E. A. Robey (2009). "Dynamic imaging of T cell-parasite interactions in the brains of mice chronically infected with *Toxoplasma gondii*." J Immunol 182(10): 6379-6393.

Simms, P. E. and T. M. Ellis (1996). "Utility of flow cytometric detection of CD69 expression as a rapid method for determining poly- and oligoclonal lymphocyte activation." Clin Diagn Lab Immunol 3(3): 301-304.

Wilson, E. H., T. H. Harris, P. Mrass, B. John, E. D. Tait, G. F. Wu, M. Pepper, E. J. Wherry, F. Dzierzinski, D. Roos, P. G. Haydon, T. M. Laufer, W. Weninger and C. A. Hunter (2009). "Behavior of parasite-specific effector CD8^+^ T cells in the brain and visualization of a kinesis-associated system of reticular fibers." Immunity 30(2): 300-311.